# Bounce: Reliable High-Dimensional Bayesian Optimization for Combinatorial and Mixed Spaces

**Leonard Papenmeier**
Lund University
leonard.papenmeier@cs.lth.se

**Luigi Nardi**
Lund University, Stanford University, DBtune
luigi.nardi@cs.lth.se

**Matthias Poloczek**
Amazon
San Francisco, CA 94105, USA
matpol@amazon.com

## Abstract

Impactful applications such as materials discovery, hardware design, neural architecture search, or portfolio optimization require optimizing high-dimensional black-box functions with mixed and combinatorial input spaces. While Bayesian optimization has recently made significant progress in solving such problems, an in-depth analysis reveals that the current state-of-the-art methods are not reliable. Their performances degrade substantially when the unknown optima of the function do not have a certain structure. To fill the need for a reliable algorithm for combinatorial and mixed spaces, this paper proposes Bounce that relies on a novel map of various variable types into nested embeddings of increasing dimensionality. Comprehensive experiments show that Bounce reliably achieves and often even improves upon state-of-the-art performance on a variety of high-dimensional problems.

## 1 Introduction

Bayesian optimization (BO) has become a 'go-to' method for optimizing expensive-to-evaluate black-box functions [27, 83] that have numerous important applications, including hyperparameter optimization for machine learning models [9, 27], portfolio optimization in finance [7], chemical engineering and materials discovery [13, 15, 26, 33, 37, 38, 40, 57, 68, 71, 81], hardware design [22, 36, 51], or scheduling problems [42]. These problems are challenging for a variety of reasons. Most importantly, they may expose hundreds of tunable parameters that allow for granular optimization of the underlying design but also lead to high-dimensional optimization tasks and the 'curses of dimensionality' [10, 63]. Typical examples are drug design [53, 76] and combinatorial testing [55]. Moreover, real-world applications often have categorical or ordinal tunable parameters, in addition to the real-valued parameters that BO has traditionally focused on [10, 27, 70]. Recent efforts have thus extended BO to combinatorial and mixed spaces. Casmopolitan of Wan et al. [79] uses trust regions (TRs) to accommodate high dimensionality, building upon prior work of Eriksson et al. [25] for continuous spaces. COMBO of Oh et al. [56] constructs a surrogate model based on a combinatorial graph representation of the function. Recently, Deshwal et al. [21] presented BODi that employs a novel type of dictionary-based embedding and showed that it outperforms the prior work. However, the causes for BODi's excellent performance are not yet well-understood and require a closer examination. Moreover, the ability of methods for mixed spaces to scale to higher dimensionalities trails behind BO for continuous domains. In particular, Papenmeier et al. [60] showed that nested embeddings allow BO to handle a thousand input dimensions, thus outperforming vanilla TR-based approaches and raising the question of whether similar performance gains are feasible for combinatorial domains.

37th Conference on Neural Information Processing Systems (NeurIPS 2023).

In this work, we assess and improve upon the state-of-the-art in combinatorial BO. In particular, we make the following contributions:

1. We conduct an in-depth analysis of two state-of-the-art algorithms for combinatorial BO, `COMBO` [56] and `BODi` [21]. The analysis reveals that their performances often degrade considerably when the optimum of the optimization problem does not exhibit a particular structure common for synthetic test problems.
2. We propose `Bounce` (**B**ayesian **o**ptimization **u**sing **in**creasingly high-dimensional **c**ombinatorial and continuous **e**mbeddings), a novel high-dimensional Bayesian optimization (HDBO) method that effectively optimizes over combinatorial, continuous, and mixed spaces. `Bounce` leverages parallel function evaluations efficiently and uses nested random embeddings to scale to high-dimensional problems.
3. We provide a comprehensive evaluation on a representative collection of combinatorial, continuous, and mixed-space benchmarks, demonstrating that `Bounce` is on par with or outperforms state-of-the-art methods.

## 2 Background and related work

**Bayesian optimization.** Bayesian optimization aims to find the global optimum $x^* \in \mathcal{X}$ of a black-box function $f : \mathcal{X} \to \mathbb{R}$, where $\mathcal{X}$ is the $D$-dimensional search space or *input space*. Throughout this paper, we consider minimization problems, i.e., we aim to find $x^* \in \mathcal{X}$ such that $f(x^*) \leq f(x)$ for all $x \in \mathcal{X}$. The search space $\mathcal{X}$ may contain variables of different types: continuous, categorical, and ordinal. We denote the number of continuous variables in $\mathcal{X}$ by $n_{\text{cont}}$ and the number of combinatorial variables by $n_{\text{comb}} = n_{\text{cat}} + n_{\text{ord}} = D - n_{\text{cont}}$, where we denote the number of categorical variables by $n_{\text{cat}}$ and the number of ordinal variables by $n_{\text{ord}}$.

**Combinatorial domains.** Extending BO to combinatorial spaces is challenging, for example, because the acquisition function is only defined at discrete locations or the dimensionality of the space grows drastically when using one-hot encoding for categorical variables. Due to its numerous applications, combinatorial BO has received increased attention in recent years. `BOCS` [6] handles the exponential explosion of combinations by only modeling lower-order interactions of combinatorial variables and imposing a sparse prior on the interaction terms. `COMBO` [56] models each variable as a graph and uses the graph-Cartesian product to represent the search space. We revisit `COMBO`'s performance on categorical problems in Appendix H.1. `CoCaBo` [67] combines multi-armed bandits and BO to allow for optimization in mixed spaces. It uses two separate kernels for continuous and combinatorial variables and proposes a weighted average of a product and a sum kernel to model mixed spaces. Liu and Wang [47] show that certain, possibly combinatorial functions can be modeled by parametric function approximators such as neural networks or random forests. As a general method to optimize the acquisition function with gradient-based methods, Daulton et al. [19] propose probabilistic reparametrization.

**High-dimensional continuous spaces.** Subspace-based methods are primarily used for continuous spaces. Wang et al. [84] propose `REMBO` for HDBO in continuous spaces using Gaussian random projection matrices. `REMBO` suffers from distortions and projections outside the search domains that the corrections of Binois et al. [11, 12] address. The `HeSBO` algorithm of Nayebi et al. [52] avoids the need for corrections by using the `CountSketch` embedding [87]. `Alebo` of Letham et al. [46] builds upon `REMBO`, learning suitable corrections of distortions. `TuRBO` [25] is a method that operates in the full-dimensional input space $\mathcal{X}$, relying on trust region (TR) to focus the search on promising regions of the search space. `BAxUS` of Papenmeier et al. [60] combines the trust region approach of `TuRBO` with the random subspace idea of `HeSBO`. `BAxUS` uses a novel family of *nested* random subspaces that exhibit better theoretical guarantees than the `CountSketch` embedding. While `BAxUS` handled a $1000D$ problem, it only considers continuous problems and cannot leverage parallel function evaluations. `GTBO` [35] assumes the existence of an axis-aligned active subspace. The algorithm first identifies "active" variables and optimizes in the full-dimensional space by placing separate strong length scale priors onto active and inactive variables. Another line of recent approaches employs Monte-Carlo tree search (MCTS) to reduce the complexity of the problem. Wang et al. [82] use MCTS to learn a partitioning of the continuous search space to focus the search on promising regions in the search space. Song et al. [74] use a similar approach, but instead of learning promising regions in the search space, they assume an axis-aligned active subspace and use MCTS to select important variables. Linear embeddings and random linear embeddings [14, 46, 52, 60, 84] require little or no training data to construct the embedding but assume a linear subspace.

**Algorithm 1** The `Bounce` algorithm

---

**Input:** initial target dimensionality $d_{\text{init}}$, evaluation budget $m$, batch size $B$, evaluation budget to input dimensionality $m_D$, # new bins added per dimension $b$, number design of experiment (DOE) points $n_{\text{init}}$

**Output:** optimizer $\boldsymbol{x}^* \in \arg\min_{\boldsymbol{x} \in \mathcal{D}} f(\boldsymbol{x})$.

1: $i \leftarrow 0, d \leftarrow d_{\text{init}}$
2: $b \leftarrow \textsc{AdjustBins}(b, d_{\text{init}}, m_D)$            ▷ Section 3
3: $m_i \leftarrow \left\lfloor \frac{b \cdot m_D \cdot d_{\text{init}}}{d_{\text{init}} \cdot (1-(b+1)^{k+1})} \right\rceil$
4: $S \leftarrow \textsc{InitialEmbedding}(d_0, n_{\text{cont}}, n_{\text{cat}}, n_{\text{comb}}, n_{\text{bin}})$      ▷ Section 3.1
5: $\mathcal{D} \leftarrow \{(\boldsymbol{z}_k, f(S^{-1}(\boldsymbol{z}_k)))\}_{k \in n_{\text{init}}}$        ▷ Sample and evaluate initial points.
6: **for** $j = 1, \ldots, m$ **do**
7:      $L_{\text{cont}} \leftarrow 0.8, L_{\text{comb}} \leftarrow \min(40, n_{\text{comb}})$
8:      **while** $L_{\text{cont}} > L_{\min}^{\text{cont}} \wedge L_{\text{comb}} > L_{\min}^{\text{comb}}$ **do**
9:          Find $B$ candidates according to Sec. 3.2
10:          Evaluate $f$ at $B$ and update $\mathcal{D}$: $\mathcal{D} \leftarrow \mathcal{D} \cup \{(\boldsymbol{z}_k, f(S^{-1}(\boldsymbol{z}_k)))\}_{k \in B}$
11:          Update $L_{\text{cont}}$ and $L_{\text{comb}}$           ▷ Section 3
12:      **if** $d < D$ **then**
13:          $i \leftarrow i + 1$        ▷ Increase index for target dimensionality.
14:          $S \leftarrow \textsc{IncreaseEmbedding}(S, b)$        ▷ Section 3.1
15:          $d \leftarrow$ # target variables in $S$
16:          $m_i \leftarrow \left\lfloor \frac{b \cdot m_D \cdot d}{d_{\text{init}} \cdot (1-(b+1)^{k+1})} \right\rceil$
17:      **else**
18:          Reset $\mathcal{D}$ by resampling and evaluating new initial points
19:          Resample $S$, reset $L_{\text{cont}}$ and $L_{\text{comb}}$, $j \leftarrow j + n_{\text{init}}$

---

**Combinatorial high-dimensional domains.** These works optimize black-box functions defined over a combinatorial or mixed space with dozens of input variables. Thebelt et al. [77] use a tree-ensemble kernel to model the Gaussian process (GP) prior and derive a formulation of the upper-confidence bound (UCB) that allows it to be optimized globally and to incorporate constraints. `RDUCB` [88] relies on random additive decompositions of the GP kernel to model the correlation between variables. `Casmopolitan` [79] follows `TuRBO` in using TRs to focus the search on promising regions of the search space and uses the Hamming distance to model TRs for combinatorial variables. For mixed spaces, `Casmopolitan` uses interleaved search and models continuous and categorical variables with two separate TRs. Kim et al. [44] use a random projection matrix to approach combinatorial problems in a continuous embedded subspace. When evaluating a point, their approach projects the continuous candidate point to the high-dimensional search space and then rounds to the next feasible combinatorial solution. Deshwal et al. [20] propose two algorithms for *permutation spaces*, which occur in problems such as compiler optimization [36] and pose special challenges due to the superexponential explosion of solutions. `BODi` [21] proposes an embedding type based on a dictionary of anchor points in the search space. The pairwise Hamming distances between the point and each anchor point $\boldsymbol{a}_i$ in the dictionary represent a point in the search space. The anchor points in the dictionary change at each iteration of the algorithm. They are sampled from the search space to cover a wide range of 'sequences', i.e., the number of changes from 0 to 1 (and vice versa) in the binary vector. The authors hypothesize that the diverse sampling procedure leads to `BODi`'s remarkable performance in combinatorial spaces with up to 60 dimensions. To our knowledge, `BODi` is the only other method combining embeddings and combinatorial spaces. We show in Section 4.6 that `BODi`'s reported good performance relies on an artificial structure of the optimizer $\boldsymbol{x}^*$ and that its performance degrades considerably when this structure is violated.

## 3 The `Bounce` algorithm

To overcome the aforementioned challenges in HDBO for real-world applications, we propose `Bounce`, a new algorithm for continuous, combinatorial, and mixed spaces. `Bounce` uses a GP [85] surrogate in a lower-dimensional subspace, the *target space*, that is realized by partitioning input variables into 'bins', the so-called *target dimensions*. `Bounce` only bins variables of the same type (categorical, binary, ordinal, and continuous). When selecting new points to evaluate, `Bounce` sets

all input variables within the same bin to a single value. It thus operates in a subspace of lower dimensionality than the input space and, in particular, maximizes the acquisition function in a subspace of lower dimensionality. `Bounce` iteratively refines its subspace embedding by splitting bins into smaller bins, allowing for a more granular optimization at the expense of higher dimensionality. Note that by splitting up bins, `Bounce` asserts that observations taken in earlier subspaces are contained in the current subspace; see Papenmeier et al. [60] for details. Thus, `Bounce` operates in a series of nested subspaces. It uses a novel TR management to leverage batch parallelism efficiently, improving over the single point acquisition of `BAxUS` [60].

**The nested subspaces.** To model the GP in low-dimensional subspaces, `Bounce` leverages `BAxUS`' family of nested random embeddings [60]. In particular, `Bounce` employs the sparse count-sketch embedding [87] in which each input dimension is assigned to exactly one target dimension. When increasing the target dimensionality, `Bounce` creates $b$ new bins for every existing bin and re-distributes the input dimensions that had previously been assigned to that bin across the now $b+1$ bins. `Bounce` allocates an individual evaluation budget $m_i$ to the current target space $\mathcal{X}_i$ that is proportional to the dimensionality of $\mathcal{X}_i$. When the budget for the current target space is depleted, and `Bounce` has not found a better solution, `Bounce` will increase the dimension of the target space until it reaches the input space of dimensionality $D$. Let $d_{\text{init}}$ denote the dimensionality of the first target space, i.e., the random embedding that `Bounce` starts with. Then `Bounce` has to increase the target dimension $\left\lceil \log_{b+1} D/d_{\text{init}} \right\rceil =: k$-times to reach the input dimensionality $D$. After calculating $k$, `Bounce` re-sets the split factor $b$ such that the distance between the predicted final target dimensionality $d_k = d_{\text{init}} \cdot (b+1)^k$ and the input dimensionality $D$ is minimized: $b = \lfloor \log_k(D/d_{\text{init}}) - 1 \rceil$, where $\lfloor x \rceil$ denotes the integer closest to $x$. This ensures that the predetermined evaluation budget for each subspace will be approximately proportional to its dimensionality. This contrasts to `BAxUS` [60] that uses a constant split factor $b$ and adjusts the initial target dimensionality $d_{\text{init}}$. The evaluation budget $m_i$ for the $i$-th subspace $\mathcal{X}_i$ is $m_i := \left\lfloor \frac{b \cdot m_D \cdot d_i}{d_{\text{init}} \cdot (1-(b+1)^{k+1})} \right\rfloor$, where $m_D$ is the budget until $D$ is reached and $b$ is the maximum number of bins added per split.

**Trust region management.** `Bounce` follows `TuRBO` [25] and `Casmopolitan` [79] in using trust regions (TRs) to efficiently optimize over target spaces of high dimensionality. TRs allow focusing on promising regions of the search space by restricting the next points to evaluate to a region centered at the current best function value [25]. TR-based methods usually expand their TR if they find better points and conversely shrink it if they fail to make progress. If the TR falls below the threshold given by the *base length*, `TuRBO` and `Casmopolitan` restart with a fresh TR elsewhere. `Casmopolitan` [79] uses different base lengths for combinatorial and continuous variables. For combinatorial variables, the distance to the currently best function value is defined in terms of the Hamming distance, and the base length is an integer. For continuous variables, `Casmopolitan` defines the base length in terms of the Euclidean distance, i.e., a real number. Similarly, `Bounce` has separate base lengths $L_{\min}^{\text{cont}}$ and $L_{\min}^{\text{comb}}$ for continuous and combinatorial variables but does not fix the factor by which the TR volume is increased or decreased upon successes or failures. Instead, the factor is adjusted dynamically so that the evaluation budget $m_i$ for the current target space $\mathcal{X}_i$ is adhered to. This design is crucial to enable batch parallelism, as we describe next.

**Batch parallelism.** We allow `Bounce` to efficiently evaluate batches of points in parallel by using a scalable TR management strategy and $q$-expected improvement ($q$EI) [5, 80, 86] as the acquisition function for batches of size $B > 1$. When `Bounce` starts with a fresh TR, we sample $n_{\text{init}}$ initial points to initialize the GP, using a Sobol sequence for continuous variables and uniformly random values for combinatorial variables.

The TR management strategy of `Bounce` differs from previous strategies [25, 60, 65, 79] in that it uses a dynamic factor to determine the TR base length. Recall that `Bounce` shrinks the TR if it fails to make progress and starts a fresh TR if the TR falls below the threshold given by the base length. `Bounce`'s rule is based on the idea that the minimum admissible TR base length should be reached when the current evaluation budget is exhausted. If one employed the strategies of `TuRBO` [25], `Casmopolitan` [79], or `BAxUS` [60] for larger batch sizes $B$ and `Bounce`'s nested subspaces, then one would spend a large part of the evaluation budget in early target spaces. For example, suppose a continuous problem, the common values for the initial, minimum, and maximum TR base length, and the constant shrinkage factor of [65]. Then, such a method has to shrink the TR base length at least seven times (i.e., evaluate $f$ $7B$-times) before it would increase the dimensionality of the target space. Thus, the method would risk depleting its budget before reaching a target space suitable

for the problem. On the other hand, we will see that Bounce chooses an evaluation budget that is smaller in low-dimensional target spaces and higher for later target spaces of higher dimensionality. Considering a 1000-dimensional problem with an evaluation budget of 1000, it uses only $3, 12,$ and $47$ samples for the first three target spaces of dimensionalities $2, 8,$ and $32$.

Bounce's strategy permits flexible TR shrinkage factors and base lengths, i.e., TR base lengths to vary within the range $[L_{\min}, L_{\max}]$. This allows Bounce to comply with the evaluation budget $m_i$ for the current target space $\mathcal{X}_i$. Suppose that Bounce has evaluated $j$ batches of $B$ points each since it last increased the dimensionality of the target space, and let $L_j$ denote the current TR base length. Observe that hence $m_i - jB$ evaluations remain for $\mathcal{X}_i$. Then Bounce sets the TR base length to $L_{j+1} := \lambda_j^{-B} L_j$ with $\lambda_j < 1$, if it found a new best point whose objective value improves upon the incumbent by at least $\varepsilon$. We call this a 'success'. Otherwise, Bounce observes a 'failure' and sets $L_{j+1} := \lambda_j^{+B} L_j$. The rationale of this rule is that if the algorithm is in iteration $j$ and only observes failures subsequently, then we apply this factor $(m_i - jB)$-times, which is the remaining number of function evaluations in the current subspace $\mathcal{X}_i$. Hence, the last batch of the $i$-th target space $\mathcal{X}_i$ will have the minimum TR base length, and Bounce will increase the target dimensionality afterward. If the TR is expanded upon a 'success', we need to adjust $\lambda_j$ not to use more than the allocated number of function evaluations in a target space. At each iteration, we therefore set adjustment factor $\lambda_j = (L_{\min}/L_j)^{1/(m_i - jB)}$. Note that $\lambda_j$ remains unchanged under this rule unless the TR expanded in the previous iteration.

**The kernel choice.** To harvest the sample efficiency of a low-dimensional target space, we would like to combine categorical variables into a single bin, even if they vary in the number of categories. This is not straightforward. For example, note that the popular one-hot encoding of categorical variables would give rise to multiple binary input dimensions, which would not be compatible with the above strategy of binning variables to form nested subspaces. Bounce overcomes these obstacles and allows variables of the same type to share a representation in the target space. We provide the details in Sect. 3.1.

For the GP model, we use the CoCaBo kernel [67]. In particular, we model the continuous and combinatorial variables with two separate $5/2-$Matérn kernels where we use automatic relevance determination (ARD) for the continuous variables and share one length scale for all combinatorial variables. Following Ru et al. [67], we use a mixture of the sum and the product kernel:

$$k(\boldsymbol{x}, \boldsymbol{x}') = \sigma_f^2 (\rho k_{\text{cmb}}(\boldsymbol{x}_{\text{cmb}}, \boldsymbol{x}'_{\text{cmb}}) k_{\text{cnt}}(\boldsymbol{x}_{\text{cnt}}, \boldsymbol{x}'_{\text{cnt}}) + (1 - \rho)(k_{\text{cmb}}(\boldsymbol{x}_{\text{cmb}}, \boldsymbol{x}'_{\text{cmb}}) + k_{\text{cnt}}(\boldsymbol{x}_{\text{cnt}}, \boldsymbol{x}'_{\text{cnt}}))),$$

where $\boldsymbol{x}_{\text{cnt}}$ and $\boldsymbol{x}_{\text{cmb}}$ are the continuous and combinatorial variables in $\boldsymbol{x}$, respectively, and $\sigma_f^2$ is the signal variance. The trade-off parameter $\rho \in [0, 1]$ is learned jointly with the other hyperparameters *via* likelihood maximization. See Appendix G.2 for additional details.

Algorithm 1 gives a high-level overview of Bounce. In Appendix A, we prove that Bounce converges to the global optimum under mild assumptions. We now explain the different components of Bounce in detail.

### 3.1 The subspace embedding of mixed spaces

Bounce supports mixed spaces of four types of input variables: categorical, ordinal, binary, and continuous variables. We discuss binary and categorical variables separately because we model them differently. The proposed embedding maps only variables of a single type to each 'bin', i.e., no target dimension of the embedding has variables of different types. Thus, target dimensions are homogeneous in this regard. Note that the number of target dimensions of each type is implied by the current bin size of the embedding that may grow during the execution. The proposed embedding can handle categorical or ordinal input variables that differ in the number of discrete values they can take.

**Continuous variables.** As common in BO, we suppose that each continuous variable takes values in a bounded interval. Thus, we may normalize each interval to $[-1, 1]$. The embedding of continuous variables, i.e., input dimensions, follows BAxUS [60]: each input dimension $D_i$ is associated with a random sign $s_i \in \{-1, +1\}$ and one or multiple input dimensions can be mapped to the same target dimension of the low-dimensional embedded subspace. Recall that Bounce works on the low-dimensional subspace and thus decides an assignment $v_j$ for every target dimension $d_j$ of the embedding. Then, all input variables mapped to this particular target dimension are set to this value $v_j$.

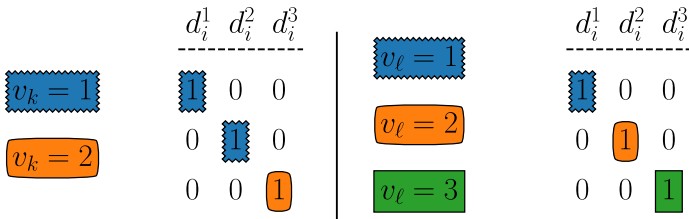

Figure 1: The mapping (or binning) of categorical and ordinal variables. Suppose that variable $v_k$ has two categories and that $v_\ell$ has three categories. Both are mapped to the target dimension $d_i$ that has cardinality $3 = \max\{2,3\}$. While the mapping of $v_\ell$ to $d_i$ is a straightforward bijection, $v_k$ has fewer categories than $d_i$. Thus, the mapping of $v_k$ to $d_i$ repeats label 1. Ordinal variables are mapped similarly.

**Binary variables.** Binary dimensions are represented by the values $-1$ and $+1$. Each input dimension $D_i$ is associated with a random sign $s_i \in \{-1, +1\}$, and the subspace embedding may map one or more binary input dimensions to the same binary target dimension. While the embedding for binary and continuous dimensions is similar, `Bounce` handles binary dimensions differently when optimizing the acquisition function.

**Categorical and ordinal variables.** Categorical variables that differ in the number of categories may be mapped to the same target dimension (bin). Suppose that the categorical variables $v_1, \ldots, v_\ell$ with cardinalities $c_1, \ldots, c_\ell$ are mapped to a single bin that is associated with the target dimension $d_j$ of the subspace embedding. Then $d_j$ is of categorical type and has $\max\{c_i \mid 1 \leq i \leq \ell\} =: c_{\max}$ distinct categories, that is, its cardinality is given by the maximum cardinality of any variable mapped to it. Thus, the bin $d_j$ can represent every category of these input variables.

Suppose that `Bounce` assigns the category $k \in \{1, \ldots, c_{\max}\}$ to the categorical bin (target dimension) $d_j$. We transform this label to a categorical assignment to each input variable $v_1, \ldots, v_\ell$, setting $v_i = \lceil k \cdot (c_i/c_{\max}) \rceil$. Recall that `Bounce` may split up bins, i.e., target dimensions, to increase the dimensionality of its subspace embedding. In such an event, every derived bin inherits the cardinality of the parent bin. This allows us to retain any observations the algorithm has taken up to this point. Analogously to the random sign for binary variables, we randomly shuffle the categories before the embedding. This reduces the risk of `Bounce` being biased towards a specific structure of the optimizer (see Appendix E).

We treat ordinal variables as categorical variables whose categories correspond to the discrete values the ordinal variable can take. For the sake of simplicity, we suppose here that an ordinal variable $v_i$ has range $\{1, 2, \ldots, c_i\}$ and $c_i \geq 2$ for all $i \in \{1, \ldots, \ell\}$. Figure 1 shows examples of the binning of categorical and ordinal variables.

### 3.2 Maximization of the acquisition function

We use expected improvement (EI) [43] for batches of size $B = 1$ and qEI [5, 80, 86] for larger batches. We optimize the EI using gradient-based methods for continuous problems and local search for combinatorial problems. We interleave gradient-based optimization and local search for functions defined over a mixed space; see Appendix G.1 for details.

## 4 Experimental evaluation

We evaluate `Bounce` empirically on various benchmarks whose inputs are combinatorial, continuous, or mixed spaces. The evaluation comprises the state-of-the-art algorithms `BODi` [21], `Casmopolitan` [79], `COMBO` [56], `SMAC` [41], and `RDUCB` [88], using code provided by the authors. We also report `Random Search` [8] as a baseline. For categorical problems, `COMBO`'s implementation suffers from a bug explained in Appendix H.2. We report the results for `COMBO` with the correct benchmark implementation as "`COMBO` (fixed)".

**The experimental setup.** We initialize every algorithm with five initial points. The plots show the performances of all algorithms averaged over 50 repetitions except `BODi`, which has 20 repetitions due to resource constraints caused by its high memory demand. The shaded regions give the standard error of the mean. We use common random seeds for all algorithms and for randomizing the

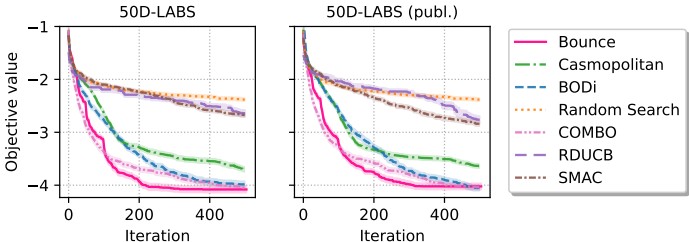

Figure 2: The 50D low-autocorrelation binary sequence problem. `Bounce` finds the best solutions, followed by `COMBO`.

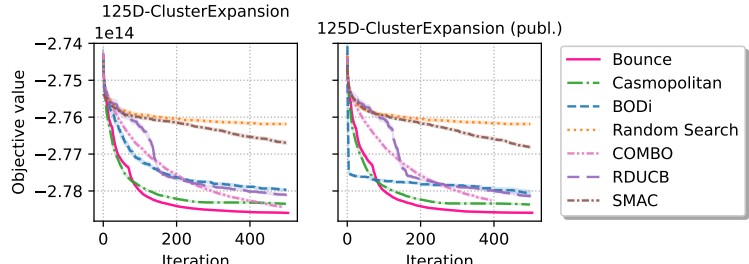

Figure 3: The 125D weighted `ClusterExpansion` maximum satisfiability problem. We plot the total negative weight of clauses. `Bounce` produces the best assignments.

benchmark functions. We run all methods for 200 function evaluations unless stated otherwise. The `Labs` (Section 4.1) and `MaxSat125` (Section 4.2) benchmarks are run for 500 evaluations.

**The benchmarks.** The evaluation uses seven established benchmarks [21]: 53D `SVM`, 50D `LABS`, 125D `ClusterExpansion` [3, 4], 60D `MaxSAT60` [21, 56], 25D `PestControl`, 53D `Ackley53`, and 25D `Contamination` [6, 39, 56]. Due to space constraints, we moved the results for the `MaxSAT60`, `Contamination`, and `Ackley53` benchmarks to Appendix B.1. For each benchmark, we report results for the originally published formulation and for a modification where we move the optimal point to a random location. The randomization procedure is fixed for each benchmark for all algorithms and repetitions. For binary problems, we flip each input variable independently with probability 0.5. For categorical problems, we randomly permute the order of the categories. We motivate this randomization in Section 4.6.

### 4.1 50D Low-Autocorrelation Binary Sequences (`LABS`)

`LABS` has $n = 50$ binary dimensions. It has important applications in communications engineering and mathematics; see [58] for details. `LABS` is a hard combinatorial problem and currently solved *via* exhaustive search. The goal is to find a sequence $x \in \{-1, +1\}^n$ with a maximum merit factor $F(x) = \frac{n^2}{2E(x)}$, where $E(x) = \sum_{k=1}^{n-1} C_k^2(x)$ and $C_k(x) = \sum_{i=1}^{n-k} x_i x_{i+k}$ for $k = 0, \ldots, n-1$ are the autocorrelations of $x$ [58]. Figure 2 summarizes the performances. We observe that `Bounce` outperforms all other algorithms on the benchmark's original and randomized versions.

### 4.2 Industrial Maximum Satisfiability: 125D `ClusterExpansion` benchmark

MaxSat is a notoriously hard problem for which various approximations and exact (exponential time) algorithms have been developed; see [32, 61] for an overview. We evaluate `Bounce` and the other algorithms on the 125-dimensional `ClusterExpansion` benchmark, a real-world MaxSAT instance with many applications in materials science [2]. Unlike the `MaxSAT60` benchmark (see Appendix B.1.3), `ClusterExpansion` is not a crafted benchmark, and its optimum has no synthetic structure [1, 3]. We treat the MaxSat problems as black-box problems; hence, algorithms do not have access to the clauses, and we cannot use the usual algorithms.

Figure 3 shows the total negative weight of the satisfied clauses as a function of evaluations. We cannot plot regret curves since the optimum is unknown [4]. We observe that `Bounce` finds better solutions than all other algorithms. `BODi` is the only algorithm for which we observe sensitivity to

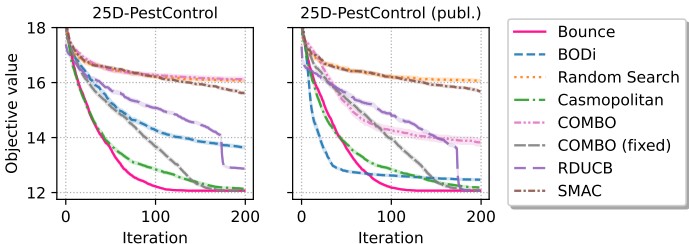

Figure 4: The 25D categorical pest control problem. `Bounce` obtains the best solutions, followed by `Casmopolitan`. `BODi`'s performance degrades significantly when shuffling the order of categories.

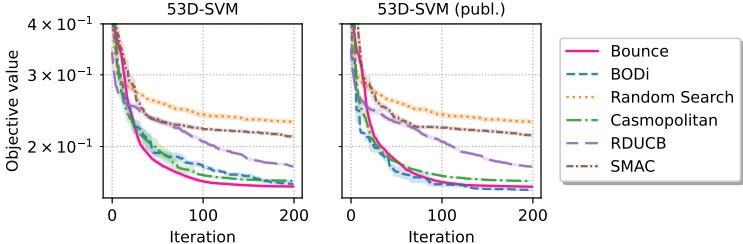

Figure 5: The 53-dimensional `SVM` benchmark. `Bounce`, `BODi`, and `Casmopolitan` achieve comparable solutions.

the location of the optimal assignment: for the published version of the benchmark, `BODi` quickly jumps to a moderately good solution but fails to make further progress.

### 4.3  25D Categorical Pest Control

`PestControl` is a more complex version of the `Contamination` benchmark and has 25 categorical variables with five categories each [56]. The task is to select one out of five actions $\{1, 2, \ldots, 5\}$ at each of 25 stations to minimize the objective function that combines total cost and a measure of the spread of the pest. We note the setting $\boldsymbol{x} = (5, 5, \ldots, 5)$ achieves a good value of 12.57, while the best value found in our evaluation is 12.07 is $\boldsymbol{x} = (5, 5, \ldots, 5, 1)$ and thus has a Hamming distance of one. The random seed used in our experiments is zero. Figure 4 summarizes the performances of the algorithms. `Bounce` is robust to the location of the global optimum and consistently obtains the best solutions. In particular, the performances of `COMBO` and `BODi` depend on whether the optimum has a certain structure. We discuss this issue in detail in Appendix H.1.

### 4.4  `SVM` – a 53D AutoML task

In the `SVM` benchmark, we optimize over a mixed space with 50 binary and 3 continuous parameters to tune an $\varepsilon$-support vector regression (SVR) model [72]. The 50 binary parameters determine whether to include or exclude an input feature from the dataset. The 3 continuous parameters correspond to the regularization parameter $C$, the kernel width $\gamma$, and the $\varepsilon$ parameter of the $\varepsilon$-SVR model [72]. Its root mean squared error on a held-out dataset gives the function value. Figure 5 summarizes the performances of the algorithms. We observe that `Bounce`, `BODi`, and `Casmopolitan` achieve comparable solutions. `BODi` performs slightly worse if the ordering of the categories is shuffled and slightly better if the optimal assignment to all binary variables is one. `COMBO` does not support continuous variables and thus was omitted.

### 4.5  `Bounce`'s efficacy for batch acquisition

We study the sample efficiency of `Bounce` when it selects a batch of $B$ points in each iteration to evaluate in parallel. Figure 6 shows the results for $B = 1, 3, 5, 10$, and 20, where `Bounce` was run for $\min(2000, 200 \cdot B)$ function evaluations. We configure `Bounce` to reach the input dimensionality after 100 evaluations for $B = 1, 3, 5$ and after $25B$ for $B = 10, 20$. We observe that `Bounce` leverages parallel function evaluations effectively: it obtains a comparable function value at a considerably smaller number of iterations, thus saving wall-clock time for applications with time-consuming function evaluations. We also studied batch acquisition for continuous problems

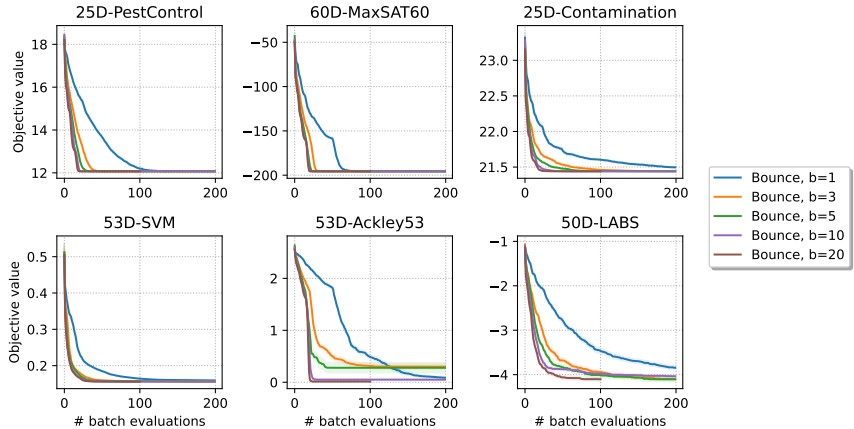

Figure 6: `Bounce` benefits from the batch acquisition that allows parallelizing function evaluations. We show the best function values obtained after each batch for batch sizes $1, 3, 5, 10$, and $20$.

and found that `Bounce` also provides significant speed-up. Due to space constraints, we deferred the discussion to Appendix C.

### 4.6 The sensitivity of `BODi` and `COMBO` to the location of the optima

The empirical evaluation reveals that the performances of `BODi` [21] and `COMBO` [56] are sensitive to the location of the optima. Both methods degrade on at least one benchmark when the optimum is moved to a randomly chosen point. This is particularly unexpected for categorical variables where moving the optimum to a random location is equivalent to shuffling the labels of the categories of each variable. Such a change of representation should not affect the performance of an algorithm.

`BODi` is more susceptible to the location of the optimizer than `COMBO`. The performance of `COMBO` degrades only on the categorical `PestControl` benchmark, whereas `BODi` degrades on five out of seven benchmarks. Looking closer, we observe that `BODi`'s performance degradation is particularly large for synthetic benchmarks like `Ackley53` and `MaxSAT60`, where setting all variables to the same value is optimal. Figure 7 summarizes the effects of moving the optimum on `BODi`. Due to space constraints, we moved the details and a discussion of categorical variables to the appendix. Similarly, setting all binary variables of the `SVM` benchmark to one produces a good objective value. It is not surprising, given that the all-one assignment corresponds to including all features previously selected for the benchmark because of their high importance.

We show in Appendix H.1 that `BODi` adds a point to its dictionary with zero Hamming distance to an all-zero or all-one solution, with a probability that increases with the dictionary size. Deshwal et al. [21, p. 7] reported that `BODi`'s performance 'tends to improve' with the size of the dictionary. Moreover, `BODi` samples a new dictionary in each iteration, eventually increasing the chance of having such a point in its dictionary. Thus, we hypothesize that `BODi` benefits from having a near-optimal solution in its dictionary, likely for all-zero or all-one solutions. For `COMBO`, Figure 4 shows that the performance on `PestControl` degrades substantially if the labels of the categories are shuffled. Then `COMBO`'s sample-efficiency becomes comparable to `Random Search`.

## 5 Discussion

BO in combinatorial spaces has many exciting and impactful applications. Its applicability to real-world problems, such as `LABS` that defy a closed-form solution, makes it a valuable tool for practitioners. Our empirical evaluation reveals that state-of-the-art methods fail to provide good solutions reliably. In particular, it finds that `BODi` and `COMBO`, which performed best in recent publications, are sensitive to the location of the optimizer. We identified design flaws in `BODi` and an implementation bug in `COMBO` as the root causes of the performance degradations.

The proposed `Bounce` algorithm is reliable for high-dimensional black-box optimization in combinatorial, continuous, and mixed spaces. The empirical evaluation demonstrates that `Bounce` reliably outperforms the state-of-the-art on a diverse set of problems. Using a novel TR management strategy, `Bounce` leverages parallel evaluations of the objective function to improve its performance. We

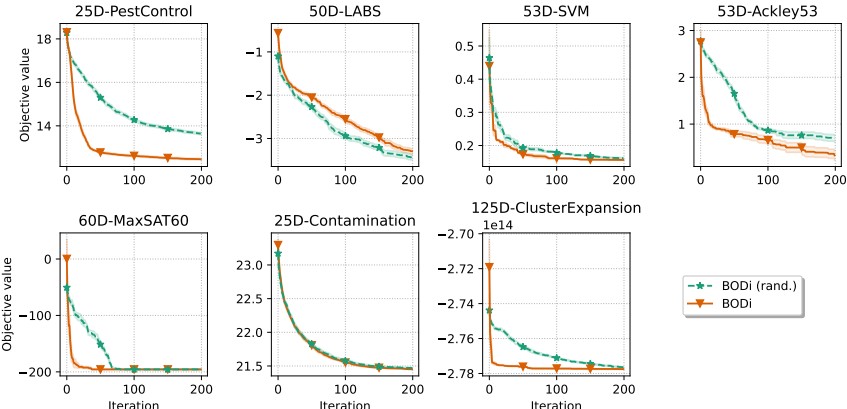

Figure 7: `BODi`'s performance degrades on five out of seven benchmarks when randomizing the location of the optimal solution: `BODi` on the default version (orange) of the benchmark and on the modified version (green, dashed) where the optimum was moved.

anticipate headroom by tailoring the modeling of combinatorial objects, e.g., arising in the search for peptides or materials discovery [28, 34, 37, 59, 75, 78]. Here it seems particularly interesting to incorporate prior belief on the importance of decision variables while maintaining the overall scalability. Moreover, extending the present work to black-box constraints [24, 30], multiple objectives, and multiple information sources [18, 31, 62] will considerably expand the applicable use cases.

**Limitations.** `Bounce` is not designed to handle noisy evaluations of the objective function. While it seems straightforward to extend `Bounce` to handle noisy evaluations, e.g., by using a Gaussian process with a noise term and acquisition functions that account for noise [70], we leave this for future work. Moreover, in applications where the categorical or ordinal variables vary substantially in the number of values they can take, there may be better ways to 'bin' them.

**Societal impact.** Bayesian optimization has recently gained wide-spread popularity for tasks in drug discovery [53], chemical engineering [15, 33, 38, 68, 71], materials discovery [28, 34, 37, 59, 75, 78], aerospace engineering [6, 45, 49], robotics [16, 17, 48, 50, 64], and many more. This highlights the Bayesian optimization community's progress toward providing a reliable 'off-the-shelf optimizer.' However, this promise is not yet fulfilled for the newer domain of mixed-variable Bayesian optimization that allows optimization over hundreds of 'tunable levers', some of which are discrete, while others are continuous. This domain is of particular relevance for the tasks above. `Bounce`'s ability to incorporate more such levers in the optimization significantly impacts the above practical applications, allowing for more granular control of a chemical reaction or a processing path, to give some examples. The empirical evaluation shows that the performance of state-of-the-art methods is highly sensitive to the location of the unknown global optima and often degenerates drastically, thus putting practitioners at risk. The proposed algorithm `Bounce`, however, achieves robust performance over a broad collection of tasks and thus will become a 'goto' optimizer for practitioners in other fields. Therefore, we open-source the `Bounce` code.[1]

## Acknowledgments and Disclosure of Funding

Leonard Papenmeier and Luigi Nardi were partially supported by the Wallenberg AI, Autonomous Systems and Software Program (WASP) funded by the Knut and Alice Wallenberg Foundation. Luigi Nardi was partially supported by the Wallenberg Launch Pad (WALP) grant Dnr 2021.0348 and by affiliate members and other supporters of the Stanford DAWN project Ant Financial, Facebook, Google, Intel, Microsoft, NEC, SAP, Teradata, and VMware. The computations were enabled by resources provided by the National Academic Infrastructure for Supercomputing in Sweden (NAISS) at the Chalmers Centre for Computational Science and Engineering (C3SE) and the National Supercomputer Centre at Linköping University, partially funded by the Swedish Research Council through grant agreement no. 2022-06725.

---

[1] `https://github.com/LeoIV/bounce`

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

# A  Consistency of `Bounce`

In this section, we prove the consistency of the `Bounce` algorithm. The proof is based on Papenmeier et al. [60] and Eriksson and Poloczek [24].

**Theorem 1** (Bounce consistency). *With the following definitions*

*Def. 1.* $(\boldsymbol{x}_k)_{k=1}^{\infty}$ *is a sequence of points of decreasing function values;*
*Def. 2.* $\boldsymbol{x}^* \in \arg\min_{\boldsymbol{x} \in \mathcal{X}}$ *is a minimizer of $f$ in $\mathcal{X}$;*

*and under the following assumptions:*

*Ass. 1.  $D$ is finite;*
*Ass. 2.  $f$ is observed without noise;*
*Ass. 3.  The range of $f$ is bounded in $\mathcal{X}$, i.e., $\exists C \in \mathbb{R}_{++}$ s.t. $|f(\boldsymbol{x})| < C \; \forall \boldsymbol{x} \in \mathcal{X}$;*
*Ass. 4.  For at least one of the minimizers $\boldsymbol{x}_i^*$ the (partial) assignment corresponding to the continuous variables lies in a (continuous) region with positive measure;*
*Ass. 5.  One `Bounce` reached the input dimensionality $D$, the continuous elements of the initial points $\{\boldsymbol{x}_{cont_i}\}_{n=1}^{n_{init}}$ after each TR restart are chosen*
*(a)  uniformly at random for continuous variables; and*
*(b)  such that every realization of the combinatorial variables has positive probability;*

*then the `Bounce` algorithm finds a global optimum with probability 1, as the number of samples $N$ goes to $\infty$.*

*Proof.* The range of $f$ is bounded per Assumption 3, and `Bounce` only considers a function evaluation a 'success' if the improvement over the current best solution exceeds a certain constant threshold. `Bounce` can only have a finite number of 'successful' evaluations because the range of $f$ is bounded per Assumption 3. For the sake of a contradiction, we suppose that `Bounce` does not obtain an optimal solution as its number of function evaluations $N \to \infty$. Thus, there must be a sequence of failures, such that the TRs in the current target space, i.e., the current subspace, will eventually reach its minimum base length. Recall that in such an event, `Bounce` increases the target dimension by splitting up the 'bins', thus creating a subspace of $(b+1)$-times higher dimensionality. Then `Bounce` creates a new TR that again experiences a sequence of failures that lead to another split, and so on. This series of events repeats until the embedded subspace eventually equals the input space and thus has dimensionality $D$. See lines $12 - 16$ in Algorithm 1 in Sect. 3.

Still supposing that `Bounce` does not find an optimum in the input space, there must be a sequence of failures such that the side length of the TR again falls below the set minimum base length, now forcing a restart of `Bounce`. Recall that at every restart, `Bounce` samples a fresh set of initial points uniformly at random from the input space; see line 18 in Algorithm 1. Therefore, with probability 1, a random sample will eventually be drawn from any subset $\mathcal{Y} \subseteq \mathcal{X}$ with positive Lebesgue measure ($\nu(\mathcal{Y}) > 0$):

$$1 - \lim_{k \to \infty} (1 - \mu(\mathcal{Y}))^k = 1, \tag{1}$$

where $\mu$ is the uniform probability measure of the sampling distribution that `Bounce` employs for initial data points upon restart [73].

Let

$$\alpha = \inf \{t : \nu\,[x \in \mathcal{X} \;\mid\; f(x) < t] > 0\}$$

denote the essential infimum of $f$ on $\mathcal{X}$ with $\nu$ being the Lebesgue measure [73].

Following Solis and Wets [73], we define the optimality region, i.e., the set of points whose function value is larger by at most $\varepsilon$ than the essential infimum:

$$R_{\varepsilon,M} = \{x \in \mathcal{X} \mid f(x) < \alpha + \varepsilon\}$$

with $\varepsilon > 0$ and $M < 0$. Because of Ass. 4, at least one optimal point lies in a region of positive measure that is continuous for the continuous variables. Therefore, we have that $\alpha = f(\boldsymbol{x}^*)$. Note that this is also the case if the domain of $f$ only consists of combinatorial variables (Ass. 5). Then, $R_{\varepsilon,M} = \{\boldsymbol{x} \in \mathcal{X} \mid f(\boldsymbol{x}) < f(\boldsymbol{x}^*) + \varepsilon\}$.

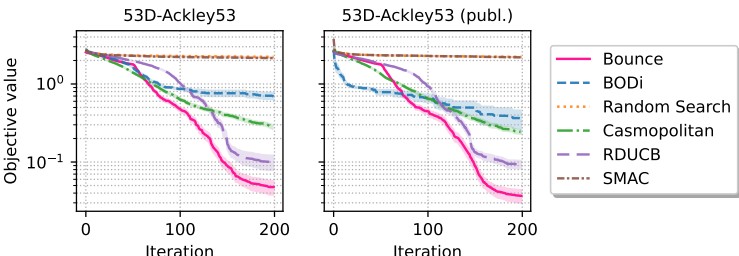

Figure 8: `Bounce` the other algorithms on the synthetic `Ackley53` benchmark function. `Bounce` outperforms all other algorithms and quickly finds excellent solutions. `BODi`'s performance degrades upon randomization.

Let $(\boldsymbol{x}_k^\star)_{k=1}^\infty$ denote the sequence of best points that `Bounce` discovers with $\boldsymbol{x}_k^\star$ being the best point up to iteration $k$. This sequence satisfies Def. 1 by construction. Note that $\boldsymbol{x}_k^\star \in R_{\varepsilon,M}$ implies that $\boldsymbol{x}_{k'}^\star \in R_{\varepsilon,M}$ for all $k' \geq k+1$ [73] because observations are noise-free. Then,

$$\mathbb{P}\left[\boldsymbol{x}_k^\star \in R_{\varepsilon,M}\right] = 1 - \mathbb{P}\left[\boldsymbol{x}_k^\star \in \mathcal{X} \setminus R_{\varepsilon,M}\right]$$
$$\geq 1 - \left(1 - \mu(R_{\varepsilon,M})\right)^k,$$

and,

$$1 \geq \lim_{k\to\infty} \mathbb{P}\left[\boldsymbol{x}_k^\star \in R_{\varepsilon,M}\right] \geq \underbrace{1 - \lim_{k\to\infty}\left(1 - \mu(R_{\varepsilon,M})\right)^k = 1}_{=1,\,\text{Eq. (1)}},$$

i.e., $\boldsymbol{x}_k^\star$ eventually falls into the optimality region [73]. By letting $\varepsilon \to 0$, $\boldsymbol{x}_k^\star$ converges to the global optimum with probability 1 as $k \to \infty$.

$\square$

## B  Additional experiments

We compare `Bounce` to the other algorithms on three additional benchmark problems: `Ackley53` and `MaxSAT60` [21]. Moreover, we run two additional studies to investigate the performance of `Bounce` further. First, we run `Bounce` on a set of continuous problems from Papenmeier et al. [60] to showcase the performance and scalability of `Bounce` on purely continuous problems. We then present a "low-sequency" version of `Bounce` to showcase how such a version can outperform its competitors on the original benchmarks by introducing a bias towards low-sequency solutions.

### B.1  `Bounce` and other algorithms on additional benchmarks

#### B.1.1  The synthetic `Ackley53` benchmark function

`Ackley53` is a 53-dimensional function with 50 binary and three continuous variables. Wan et al. [79] discretized 50 continuous variables of the original Ackley function, requiring these variables to be either zero or one. This benchmark was designed such that the optimal value of $0.0$ is at the origin $\boldsymbol{x} = (0, \dots, 0)$. Here, we perturb the optimal assignment of combinatorial variables by flipping each binary variable with probability $1/2$. Figure 8 summarizes the performances of the algorithms. `Bounce` outperforms all other algorithms and proves to be robust to the location of the optimum point. `Casmopolitan` is a distanced runner-up. `BODi` initially outperforms `Casmopolitan` on the published benchmark version but falls behind later.

#### B.1.2  Contamination control

The `Contamination` benchmark models a supply chain with 25 stages [39]. At each stage, a binary decision is made whether to quarantine food that has not yet been contaminated. Each such intervention is costly, and the goal is to minimize the number of contaminated products and prevention cost [6, 56]. Figure 9 shows the performances of the algorithms.

`Bounce`, `Casmopolitan`, and `BODi` all produce solutions of comparable objective value. `Bounce` and `Casmopolitan` find better solutions than `BODi` initially, but after about 100 function evaluations, the solutions obtained by the three algorithms are typically on par.

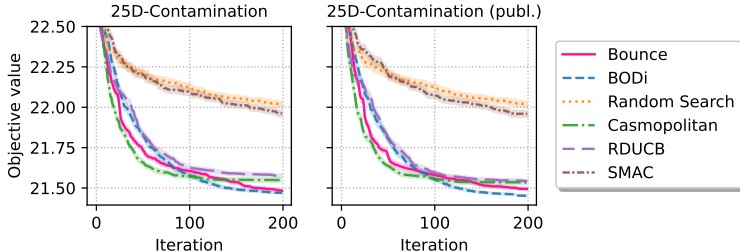

Figure 9: `Bounce` and the other algorithms on the 25-dimensional contamination problem. `Bounce` performs on par with `Casmopolitan` and `BODi` on both versions of the benchmark.

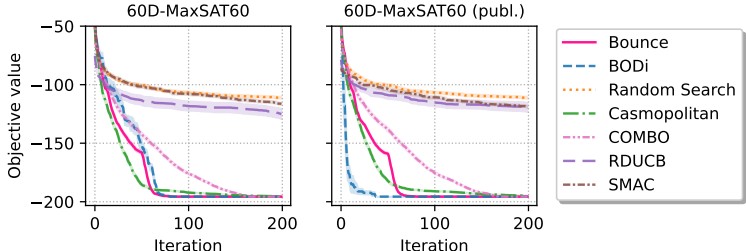

Figure 10: `Bounce` and other algorithms on the 60-dimensional weighted maximum satisfiability problem. `Bounce` is the first to find an optimal solution (left). On the published version (right), `Bounce` comes in second after `BODi`.

### B.1.3  The `MaxSAT60` benchmark

`MaxSAT60` is a 60-dimensional, weighted instance of the Maximum Satisfiability (MaxSAT) problem. MaxSAT is a notoriously hard combinatorial problem that cannot be solved in polynomial time (unless $\mathbb{P} = \mathbb{NP}$). The goal is to find a binary assignment to the variables that satisfies clauses of maximum total weight. For every $i$ in $\{1, 2, \ldots, d\}$, this benchmark has one clause of the form $x_i$ with a weight of 1 and 638 clauses of the form $\neg x_i \lor \neg x_j$ with a weight of 61. Following [21, 56, 79], we normalize these weights to have zero mean and unit standard deviation. This normalization causes the one-variable clauses to have a *negative weight*, i.e., the function value improves if such a clause is not satisfied, which is atypical behavior for a MaxSAT problem. Since the clauses with two variables are satisfied for $x_i = x_j = 0$ and the clauses with one variable of negative weights are never satisfied for $x_i = 0$, the normalized benchmark version has a global optimum at $\boldsymbol{x}^* = (0, \ldots, 0)$ by construction. The problem's difficulty is finding an assignment for variables such that *all* two-variable clauses are satisfied and *as many* one-variable clauses as possible are not captured by normalized weights.

Figure 10 summarizes the performances of the algorithms. The general version that attains the global optimum for a randomly selected binary assignment is shown on the left. The special case where the global optimum is set to the all-zero assignment is shown on the right.

We observe that `Bounce` requires the smallest number of samples to find an optimal assignment in general, followed by `BODi` and `Casmopolitan`. Only in the special case where the optimum is the all-zero assignment, `BODi` ranks first, confirming the corresponding result in Deshwal et al. [21].

# C An evaluation of `Bounce` on continuous problems

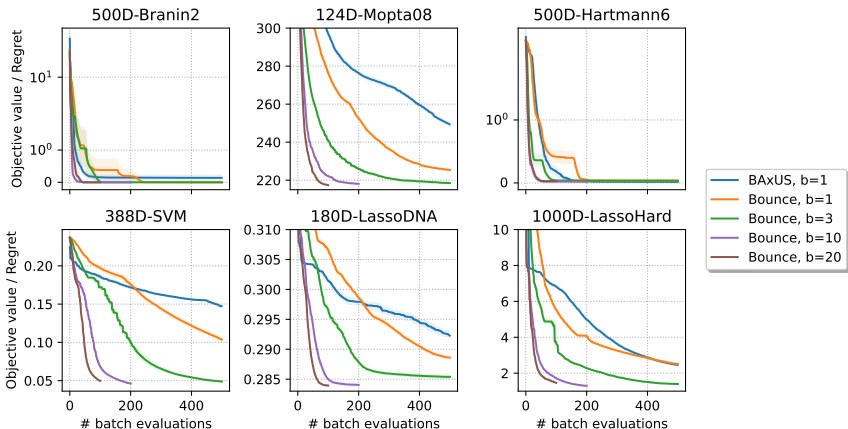

Figure 11: `Bounce` on continuous problems with different batch sizes (plotted in terms of batch evaluations).

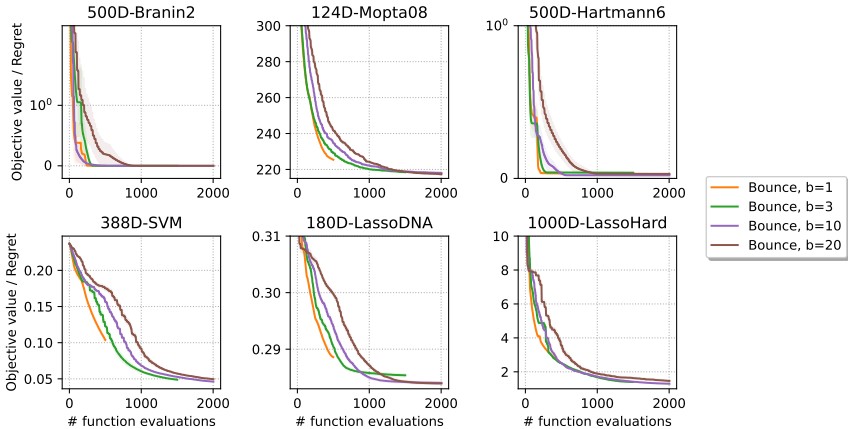

Figure 12: `Bounce` on continuous problems with different batch sizes (plotted in terms of function evaluations).

To showcase the performance and scalability of `Bounce`, we run it on a set of continuous problems from Papenmeier et al. [60]. The 124-dimensional `Mopta08` benchmark is a constrained vehicle optimization problem. We adopt the soft-constrained version from Eriksson and Jankowiak [23]. The 388-dimensional `SVM` problem [23] concerns the classification performance with an SVR on the slice localization dataset. The 180-dimensional `LassoDNA` benchmark [69] is a sparse regression problem on a real-world dataset, and the 1000-dimensional `LassoHard` benchmark optimizes over a synthetic dataset. The 500-dimensional `Branin2` and `Hartmann6` problems are versions of the 2- and 6-dimensional benchmark problems where additional dimensions with no effect on the function value were added.

We set the number of function evaluations to $\max(2000, 500B)$ for a batch size of $B$ and configure `Bounce` such that it reaches the input dimensionality after 500 function evaluations. Figures 11 and 12 show the simple regret for the synthetic `Branin2` and `Hartmann6` problems, and the best function value obtained after a given number of batch (Figure 11) or function (Figure 12) evaluations for the remaining problems: `Mopta08`, `SVM`, `LassoDNA`, and `LassoHard`.

We observe that `Bounce` always benefits from more parallel function evaluations. The difference between smaller batch sizes, such as $B = 1$ and $B = 3$ or $B = 3$ and $B = 10$, is more remarkable than between larger batch sizes, like $B = 10$ and $B = 20$. Parallel function evaluations prove

especially effective on `SVM` and `LassoDNA`. Here, the optimization performance improves drastically. We conclude that a small number of parallel function evaluations already helps to considerably increase the optimization performance.

On the synthetic `Branin2` and `Hartmann6` problems, `Bounce` quickly converges to the global optimum. Here, we see that a larger number of parallel function evaluations also helps in converging to a better solution.

In Figure 11, we also compare to `BAxUS` by Papenmeier et al. [60], which does not support parallel function evaluations and is therefore run with a batch size of 1. We observe that `Bounce` with a batch size of 1 outperforms `BAxUS` on all benchmarks except for `Hartmann6`, showcasing `Bounce`'s overall performance improvements even for continuous problems and single-element batches.

## D    Batched evaluations on mixed and combinatorial problems

In addition to Figure 6, which shows the best objective value in relation to the number of batches, Figure 13 shows the objective value in terms of function evaluations.

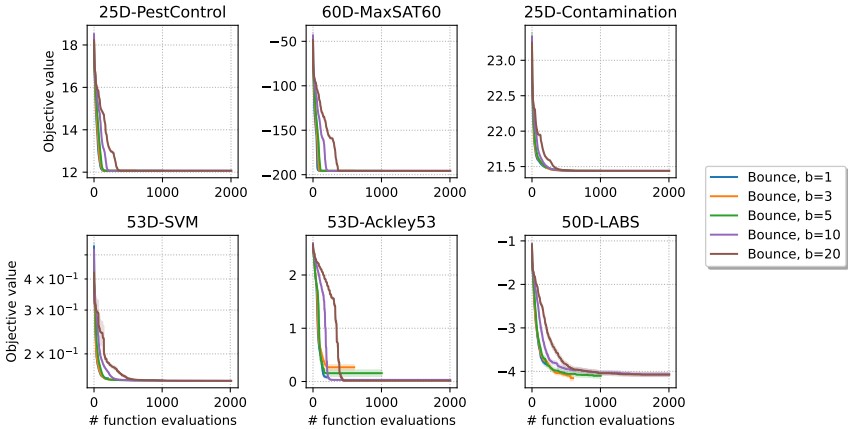

Figure 13: `Bounce` on mixed and combinatorial problems with different batch sizes (plotted in terms of function evaluations).

The figure confirms that `Bounce` leverages parallel function values efficiently.

## E    Low-sequency version of `Bounce`

We show how we can bias `Bounce` towards low-sequency solutions. A binary vector has low sequency if there are few changes from 0 to 1 or vice versa. Similarly, a solution to a categorical or ordinal problem has low sequency if there are few changes from one category or level to another. We remove the random signs (for binary and continuous variables) and the random offsets (for categorical and ordinal variables) from the `Bounce` embedding. We conduct this study to show a) that `Bounce` can outperform `BODi` on the unmodified versions of the benchmark problems if we introduce a similar bias towards low-sequency solutions and b) that the random signs empirically show to remove biases towards low-sequency solutions. However, we want to emphasize that the results of this section are not representative of the performance of `Bounce` on arbitrary real-world problems. Nevertheless, if one knows that the problem has a low-sequency structure, then `Bounce` can be configured to exploit this structure and outperform `BODi`.

Figure 14 shows the results of the low-sequency version of `Bounce` on the original benchmarks from Section 4. We observe that `Bounce` outperforms `BODi` and the other algorithms on the unmodified versions of the benchmark problems. This shows that `Bounce` can outperform `BODi` on the unmodified version of the benchmarks if we introduce a similar bias towards low-sequency solutions.

Figure 15 shows the results of the low-sequency version of `Bounce` on the flipped benchmarks from Section 4. The low-sequency version of `Bounce` is robust towards the randomization of the optimal point.

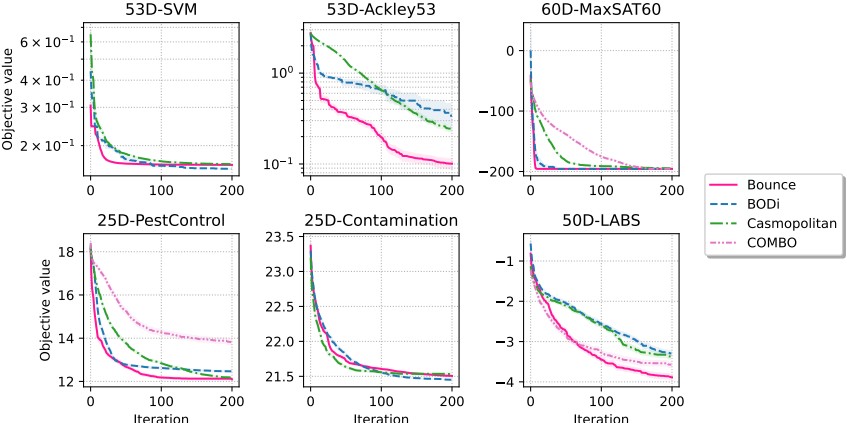

Figure 14: 'Low-sequency' version of `Bounce` on the **original** benchmarks from Section 4: with a bias towards low-sequency solutions, `Bounce` outperforms `BODi` on the original versions of the benchmark problems.

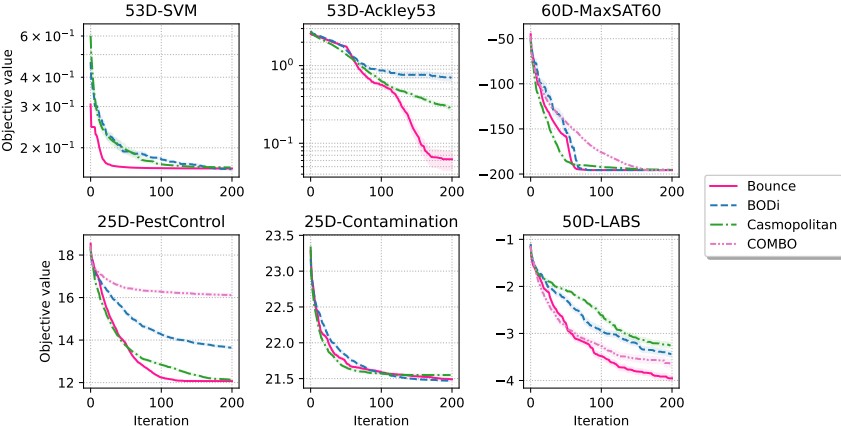

Figure 15: "Low-sequency" version of `Bounce` on the **modified** benchmarks from Section 4.

## F  Effect of trust-region management

`Bounce` uses a novel trust-region management that differs from previous approaches in that it allows arbitrary trust region base lengths in $[L_{\min}, L_{\max}]$. This strategy allows `Bounce` to efficiently leverage parallel function evaluations, which we refer to as batch acquisition.

We compare `Bounce`'s trust-region management strategy with the strategy employed by `BAxUS` [60] for purely continuous benchmarks and an adapted version of `Casmopolitan`'s [79] strategy for mixed or discrete-space benchmarks. For the latter, the difference is that in `Bounce` we reduce the failure tolerance as described by Papenmeier et al. [60]: We first calculate the number of times the TR base length needs to be reduced to reach the minimum TR base length as $k = \left\lfloor \log_{1.5^{-1}} \frac{L_{\min}}{L_{\text{init}}} \right\rfloor$, and then find the failure tolerance for the $i$-th target space by $\tau_{\text{fail}}^i = \max\left(1, \min\left(\left\lceil \frac{m_i^s}{k} \right\rceil\right)\right)$, where $m_i^s$ is the budget for the $i$-th target space [60]. This change in regard to `Casmopolitan` is necessary because `Casmopolitan`'s failure tolerance of $40$ [79] would cause the algorithm to spend a large part of the evaluation budget in initial target spaces of low dimensionality.

We show the effect of this strategy on discrete and mixed-space benchmarks in Figure 16. Figure 17 shows the effect on continuous benchmarks. All experiments are replicated 50 times. We observe that the proposed trust-region management method not only enables efficient batch parallelism but also improves the performance for single-function evaluations when compared to the respective baseline for the TR management that we stated above. The TR management proposed here usually

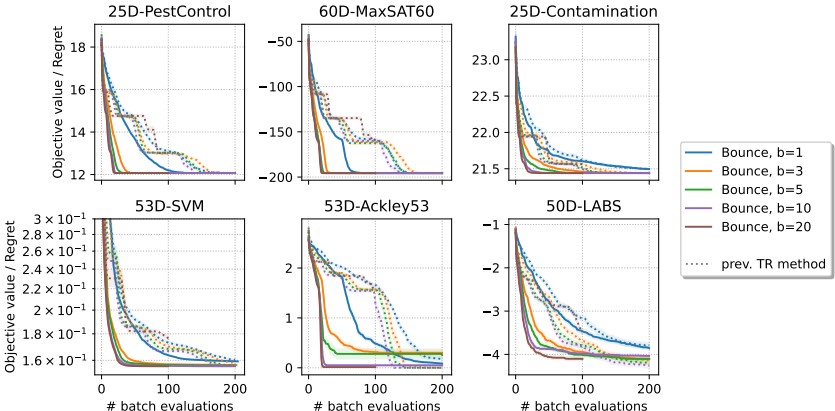

Figure 16: Effect of the proposed trust region management on the discrete and mixed benchmarks. Here, the baseline is the TR management of `Casmopolitan`. We see that often, even large batch sizes for the old TR management strategy do not outperform the proposed new strategy with a batch size of 1.

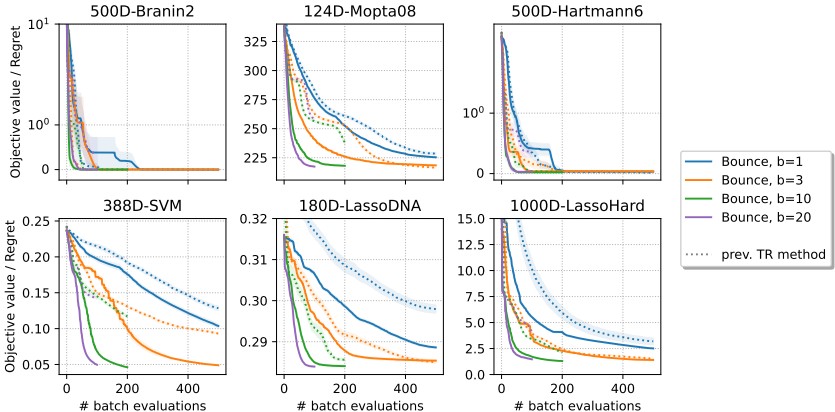

Figure 17: Effect of the proposed trust-region management on the continuous benchmarks. Here, the baseline is the TR management of `BAxUS`. We observe that larger batches help to improve performance for both methods. The new strategy, however, outperforms the baseline in almost all cases with the exception of `LassoDNA` with a batch size of 3.

provides better solutions at the same number of batches than the respective baseline TR management. There are a few exceptions. On `Labs`, the previous TR management strategy outperforms `Bounce`'s strategy by a small margin for batch sizes 5, 10, and 20, and on `Ackley53`, the previous strategy converges to a better solution for a batch size of three, five, and ten.

On the continuous benchmarks (Figure 17), we observe that the new TR management strategy also improves the performance for single function evaluations with the exception of `Branin2`. Similar to the mixed and combinatorial-space benchmarks, large batch sizes (10 and 20) bring little advantage compared to a batch size of 5. For `Hartmann6`, the previous strategy with a batch size of 20 performs worse than 3.

# G   Implementation details

We implement `Bounce` in Python using the `BoTorch` [5] and `GPyTorch` [29] libraries.

We employ a $\Gamma(1.5, 0.1)$ prior on the lengthscales of both kernels and a $\Gamma(1.5, 0.5)$ prior on the signal variance. We further use a $\Gamma(1.1, 0.1)$ prior on the noise variance.

Motivated by Wan et al. [79] and Eriksson et al. [25], we use an initial trust region baselength of 40 for the combinatorial variables, and 0.8 for the continuous variables. We maintain two separate TR shrinkage and expansion parameters ($\gamma_{\text{cmb}}$ and $\gamma_{\text{cnt}}$) for the combinatorial and continuous variables, respectively such that each TR base length reaches its respective minimum of 1 and $2^{-7}$ after a given number of function evaluations. When `Bounce` finds a better or worse solution, we increase or decrease both TR base lengths.

We use the author's implementations for `COMBO`[2], `BODi`[3], `RDUCB`[4], `SMAC`[5], and `Casmopolitan`[6]. We use the same settings as the authors for COMBO, RDUCB, SMAC, and BODi. For `Casmopolitan`, we use the same settings as the authors for benchmarks reported in Wan et al. [79] and set the initial trust region base length to 40 otherwise.

Due to its high-memory footprint, we ran `BODi` on NVidia A100 80GB GPUs for 300 GPU/h. We ran `Bounce` on NVidia A40 GPUs for 2,000 GPU/h. We ran the remaining methods for 20,000 GPU/h on one core of Intel Xeon Gold 6130 CPUs with 60GB of memory.

### G.1   Optimization of the acquisition function

We use different strategies to optimize the acquisition function depending on the type of variables present in a problem.

**Continuous problems.**   For purely continuous problems, we follow a similar approach as `TuRBO` [25]. In particular, we use the lengthscales of the GP posterior to shape the TR. We use gradient descent to optimize the acquisition function within the TR bounds with 10 random restarts and 512 raw samples. For a batch size of 1, we use analytical EI. For larger batch sizes, we use the `BoTorch` implementation of qEI [5, 66, 86].

**Binary problems.**   Similar to Wan et al. [79], we use discrete TRs centered on the current best solution. A discrete TR describes all solutions with a certain Hamming distance to the current best solution.

We use a local search approach to optimize the acquisition function for all problems with a combinatorial search space of only binary variables: When starting the optimization, we first create a set of $\min(5000, \max(2000, 200 \cdot d_i))$ random solutions. The choice of the number of random solutions is based on `TuRBO` [25]. For each candidate, we first draw $L_i$ indices uniformly at random from $\{1, \ldots, d_i\}$ without replacement, where $L_i$ is the TR length at the $i$-th iteration. We then sample $d_i$ values in $\{0, 1\}$ and set the candidate at the sampled indices to the sampled values. All other values are set to the values of the current best solution. Note that this construction ensures that each candidate solution lies in the TR bounds of the current best solution. We add all neighbors (i.e., points with a Hamming distance of 1) of the current best solution to the set of candidates. This is inspired by `BODi` [21]. We find the 20 candidates with the highest acquisition function value and use local search to optimize the acquisition function within the TR bounds: At each local search step, we create all direct neighbors that do not coincide with the current best solution or would violate the TR bounds. We then move the current best solution to the neighbor with the highest acquisition function value. We repeat this process until the acquisition function value does not increase anymore. Finally, we return the best solution found during local search.

**Categorical problems.**   We adopt the approach for binary problems, i.e., we first create a set of random solutions with the same size as for purely binary problems and start the local search on the 20 best initial candidates.

Suppose the number of categorical variables of the problem is smaller or equal to the current TR length. In that case, we sample, for each candidate and each categorical variable, an index uniformly at random from $\{1, 2, \ldots, |v_i|\}$ where $|v_i|$ is the number of values of the $i$-th categorical variable. We then set the candidate at the sampled index to 1 and all other values to 0.

---

[2]`https://github.com/QUVA-Lab/combo`, unspecified license, last access: 2023-05-04

[3]`https://github.com/aryandeshwal/bodi`, no license provided, last access: 2023-05-04

[4]`https://github.com/huawei-noah/HEBO/tree/master/RDUCB`, MIT license, last access: 2023-10-20

[5]`https://github.com/automl/pysmac`, AGPL-3.0 license, last access 2023-10-20

[6]`https://github.com/xingchenwan/casmo`, MIT license, last access: 2023-05-04

If the number of categorical variables of the problem is larger than the current TR length $L_i$, we first sample $L_i$ categorical variables uniformly at random from $[d_i]$ without replacement. For each initial candidate and each sampled categorical variable, we sample an index uniformly at random, for which we set the categorical variable to 1 and all other values to 0. The values for the variables that were not sampled are set to the values of the current best solution.

As for the binary case, we add all neighbors of the current best solution to the set of candidates, and we sample the 20 candidates with the highest acquisition function value.

We then use multi-start local search to optimize the acquisition function within the TR bounds while neighbors are created by changing the index of one categorical variable. Again, we repeat until convergence and return the best solution found during the local search.

**Ordinal problems.** The construction for ordinal problems is similar to the one for categorical problems.

Suppose the number of ordinal variables of the problem is smaller or equal to the current TR length. In that case, we sample an ordinal value uniformly at random to set the ordinal variable for each candidate and each ordinal variable. Otherwise, we choose as many ordinal variables as each candidate's current TR length and sample an ordinal value uniformly at random to set the ordinal variable. We add all neighbors of the current best solution, all solutions where the distance to the current best solution is 1 for one ordinal variable, to the set of candidates. We then sample the 20 candidates with the highest acquisition function value and use local search to optimize the acquisition function within the TR bounds. In the local search, we increment or decrement the value of a single ordinal variable.

**Mixed problems.** Mixed problems are effectively handled by treating every variable type separately. Again, we create a set of initial random solutions where the values for the different variable types are sampled according to the abovementioned approaches. This can lead to solutions outside the TR bounds. We remove these solutions and find the 20 best candidates only across the solutions within the TR bounds.

When optimizing the acquisition function, we differentiate between continuous and combinatorial variables. We optimize the continuous variables by gradient descent with the same settings as purely continuous problems. When optimizing, we fix the values for the combinatorial values.

We use local search to optimize the acquisition function for the combinatorial variables. In this step, we fix the values for the continuous variables and only optimize the combinatorial variables. We create the neighbors by creating neighbors within Hamming distance of 1 for each combinatorial variable type and then combining these neighbors. Again, we run a local search until convergence.

We do five interleaved steps, starting with the continuous variables and ending with the combinatorial variables.

### G.2 Kernel choice

We use the `CoCaBo` kernel [67] with one global lengthscale for the combinatorial and ARD for the continuous variables:

$$k(\boldsymbol{x}, \boldsymbol{x}') = \sigma_f^2 (\rho k_{\text{cmb}}(\boldsymbol{x}_{\text{cmb}}, \boldsymbol{x}'_{\text{cmb}}) k_{\text{cnt}}(\boldsymbol{x}_{\text{cnt}}, \boldsymbol{x}'_{\text{cnt}})$$
$$+ (1 - \rho)(k_{\text{cmb}}(\boldsymbol{x}_{\text{cmb}}, \boldsymbol{x}'_{\text{cmb}}) + k_{\text{cnt}}(\boldsymbol{x}_{\text{cnt}}, \boldsymbol{x}'_{\text{cnt}})))$$
$$k_{\text{cmb}}(\boldsymbol{x}_{\text{cmb}}, \boldsymbol{x}'_{\text{cmb}}) = \left(1 + \frac{\sqrt{5} r_{\text{cmb}}}{\ell_{\text{cmb}}} + \frac{5 r_{\text{cmb}}^2}{3 \ell_{\text{cmb}}^2}\right) \exp\left(-\frac{\sqrt{5} r_{\text{cmb}}}{\ell_{\text{cmb}}}\right)$$
$$r_{\text{cmb}} = \|\boldsymbol{x}_{\text{cmb}} - \boldsymbol{x}'_{\text{cmb}}\|$$
$$k_{\text{cnt}}(\boldsymbol{x}_{\text{cnt}}, \boldsymbol{x}'_{\text{cnt}}) = \left(1 + \sqrt{5} r_{\text{cnt}} + 5 r_{\text{cnt}}^2\right) \exp\left(-\sqrt{5} r_{\text{cnt}}\right)$$
$$r_{\text{cnt}} = \sqrt{\sum_{i \in [d];\, d_i \text{ continuous}} \frac{(x_i - x'_i)^2}{\ell_{\text{cnt},i}^2}}$$

where $\boldsymbol{x}_{\text{cnt}}$ and $\boldsymbol{x}_{\text{cmb}}$ are the continuous and combinatorial variables in $\boldsymbol{x}$, respectively, i.e.,

$$\boldsymbol{x}_{\text{cmb}} = (x_i : \ d_i \text{ is combinatorial})$$

$$x_{\text{cnt}} = (x_i : d_i \text{ is continuous})$$

and $\rho \in [0, 1]$ is a tradeoff parameter learned jointly with the other hyperparameters during the likelihood maximization. Here, $\ell_{\text{cmb}}$ and $\ell_{\text{cnt},i}$ are the lengthscale hyperparameters.

## H  Additional analysis of `BODi` and `COMBO`

### H.1  Analysis of `BODi`

**Binary problems.** `BODi` by Deshwal et al. [21] uses a dictionary of anchor points $\boldsymbol{A} = (\boldsymbol{a}_1, \ldots, \boldsymbol{a}_m)$ to encode a candidate point $\boldsymbol{z}$. In particular, the $i$-th entry of the $m$-dimensional embedding $\phi_{\boldsymbol{A}}(\boldsymbol{z})$ is obtained by computing the Hamming-distance between $\boldsymbol{z}$ and $\boldsymbol{a}_i$. Notably, Deshwal et al. [21] choose the dimensionality of the embedding $m$ as 128, which is larger than the dimensionality of the objective functions themselves.

The sampling procedure for dictionary elements $\boldsymbol{a}_i$ is chosen to yield a wide range of *sequencies*. The sequency of a binary string is defined as the number of times the string changes from 0 to 1 and vice versa. Deshwal et al. [21] propose two approaches to generate the dictionary elements: (i) using binary wavelets, and (ii) by first drawing a Bernoulli parameter $\theta_i \sim \mathcal{U}(0, 1)$ for each $i \in [m]$ and then drawing a binary string $\boldsymbol{a}_i$ from the distribution $\mathcal{B}(\theta_i)$. The latter approach is their default method.

We prove that `BODi` generates an all-zeros (or, by a similar symmetry argument, all-ones) representer point with a probability that is significantly higher than $2^{-D}$. Moreover, we claim without proof that a similarly increased probability holds for points with low Hamming distance to all-zeros or all-ones. This is consistent with the intention of `BODi` to sample points of diverse sequency and not necessarily an issue.

However, we claim without proof that having such points in the dictionary substantially increases the probability that `BODi` evaluates the all-zeros (and, by symmetry, the all-ones) points. That hypothesis is consistent with our observation in Section 4.6 that `BODi` has a much higher chance of finding good or optimal solutions when they are near the all-zero point.

Deshwal et al. [21] choose the dictionary to have $m = 128$ dictionary elements. Given a Bernoulli parameter $\theta_i$, the probability that the $i$-th dictionary point $\boldsymbol{a}_i$ is a point of sequency zero is given by $\theta_i^D + (1 - \theta_i)^D$:

$$\mathbb{P}(\text{``zero sequency''} \mid \theta_i) = \prod_{i=1}^{m} \theta_i^D + (1 - \theta_i)^D$$

Then, since $\theta_i$ follows a uniform distribution, the overall probability for a point of zero sequency is given by

$$\mathbb{P}(\text{``zero sequency''}) = 1 - \underbrace{\prod_{i=1}^{m} \int_0^1 \left(1 - \theta_i^D - (1 - \theta_i)^D\right) \underbrace{p(\theta_i)}_{=1} d\theta_i}_{\text{prob. of } m \text{ times not zero sequency}}$$

$$= 1 - \prod_{i=1}^{m} \left(\theta_i - \frac{\theta_i^{D+1}}{D+1} + \frac{(1 - \theta_i)^{D+1}}{D+1}\right)\Bigg|_{\theta_i = 1}$$

$$= 1 - \left(1 - \frac{2}{D+1}\right)^m,$$

i.e., the probability of at least one dictionary element being of sequency zero is $1 - \left(1 - \frac{2}{D+1}\right)^m$. The probability of `BODi`'s dictionary to contain a zero-sequency point increases with the number of dictionary elements $m$ and decreases with the function dimensionality $D$ (see Figure 18).

For instance, for the 60-dimensional `MaxSAT60` benchmark, the probability that at least one dictionary element is of sequency zero is $1 - \left(1 - \frac{2}{60+1}\right)^{128} \approx 0.986$ (see Figure 18).

Note that at least one point $\boldsymbol{z}^*$ has a probability of $\leq 1/2^d$ to be drawn. The probability of the dictionary containing that $\boldsymbol{z}^*$ is less than or equal to $1 - \left(1 - 1/2^d\right)^m$ which is already less than

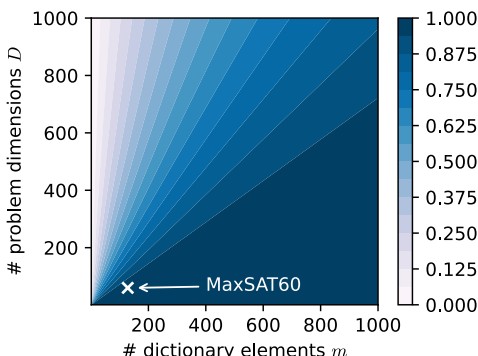

Figure 18: Probabilities of `BODi` to contain a zero-sequence solution for different choices of the dictionary size $m$ and the function dimensionality $D$.

0.01 for $d = 14$ and $m = 128$. In Section 4, we show that randomizing the optimal point structure leads to performance degradation for `BODi`. We hypothesize this is due to the reduced probability of the dictionary containing the optimal point after randomization.

**Categorical problems.** We calculate the probability that `BODi` contains a vector in its dictionary where all elements are the same. For categorical problems, `BODi` first samples a vector $\boldsymbol{\theta}$ from the $\tau_{\max}$-simplex $\Delta^{\tau_{\max}}$ for each vector $\boldsymbol{a}_i$ in the dictionary, with $\tau_{\max}$ being the maximum number of categories across all categorical variables of a problem. We assume that all variables have the same number of categories as is the case for the benchmarks in Deshwal et al. [21]. Let $\tau$ be the number of categories of the variables. For each element in $\boldsymbol{a}_i$, `BODi` draws a value from the categorical distribution with probabilities $\boldsymbol{\theta}$. While line 7 in Algorithm 5 in Deshwal et al. [21] might suggest that the elements in $\boldsymbol{\theta}$ are shuffled for every element in $\boldsymbol{a}_i$, we observe that $\boldsymbol{\theta}$ remains fixed based on the implementation provided by the authors[7]. The random resampling of elements from $\boldsymbol{\theta}$ is probably only used for benchmarks where the number of realizations differs between categorical variables.

Then, for a fixed $\boldsymbol{\theta}$, the probability that all $D$ elements in $\boldsymbol{a}_i$ for any $i$ are equal to some fixed value $t \in \{1, 2, \ldots, \tau\}$ is given by $\theta_t^d$. The probability that, for any of the $m$ dictionary elements, all $D$ elements in $\boldsymbol{a}_i$ are equal to some fixed value $t \in \{1, 2, \ldots, \tau\}$ is given by

$$\mathbb{P}(\text{``all one specific category''}) = 1 - \prod_{i=1}^{m} \int (1 - \theta_t^D) p(\theta_t) d\theta_t. \tag{2}$$

We note that $\boldsymbol{\theta}$ follows a Dirichlet distribution with $\boldsymbol{\alpha} = \mathbf{1}$ [54]. Then, $\theta_t$ is marginally $\text{Beta}(1, \tau-1)$-distributed [54]. With that, Eq. (2) becomes

$$\mathbb{P}(\text{``all one specific category''}) = 1 - \prod_{i=1}^{m} \mathbb{E}_{\theta_t \sim \text{Beta}(1, \tau-1)} \left[ 1 - \theta_t^D \right]$$

$$= 1 - \prod_{i=1}^{m} 1 - \mathbb{E}_{\theta_t \sim \text{Beta}(1, \tau-1)} \left[ \theta_t^D \right]$$

now, by using the equality $\mathbb{E}[x^D] = \prod_{r=0}^{D-1} \frac{\alpha+r}{\alpha+\beta+r}$ for the $D$-th raw moment of a $\text{Beta}(\alpha, \beta)$ distribution [54]

$$= 1 - \left( 1 - \prod_{r=0}^{D-1} \frac{1+r}{\tau+r} \right)^m$$

$$= 1 - \left( 1 - \frac{1}{\tau} \cdot \frac{2}{\tau+1} \cdot \ldots \cdot \frac{D}{\tau+D-1} \right)^m$$

---

[7]See `https://github.com/aryandeshwal/bodi/blob/aa507d34a96407b647bf808375b5e162ddf10664/bodi/categorical_dictionary_kernel.py#L18`

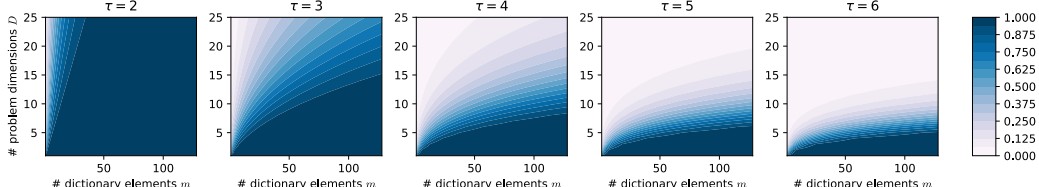

Figure 19: Probabilities of `BODi`'s dictionary to contain at least one categorical point where each category has the same value. The probability increases with the number of dictionary elements $m$ but decreases with the number of categories $\tau$ and the number of problem dimensions $D$.

$$= 1 - \left( 1 - \frac{D! \, \tau!}{(\tau + D - 1)!} \right)^m$$

We discussed in Section 4.3 that the `PestControl` benchmark obtains a good solution at $\boldsymbol{x} = \boldsymbol{5}$. One could assume that `BODi` performs well on this benchmark because its dictionary will likely contain this point. However, we observe that the probability is effectively zero for $\tau = 5$, $m = 128$, and $D = 25$ (see Figure 19), which are the choices for the `PestControl` benchmark in Deshwal et al. [21]. This raises the question of (i) whether our hypothesis is wrong and (ii) what the reason for `BODi`'s performance degradation on the `PestControl` benchmark is.

We show that `BODi`'s reference implementation differs from the algorithmic description in an important detail, causing `BODi` to be considerably more likely to sample category five on `PestControl` (or the "last" category for arbitrary benchmarks) than any other category.

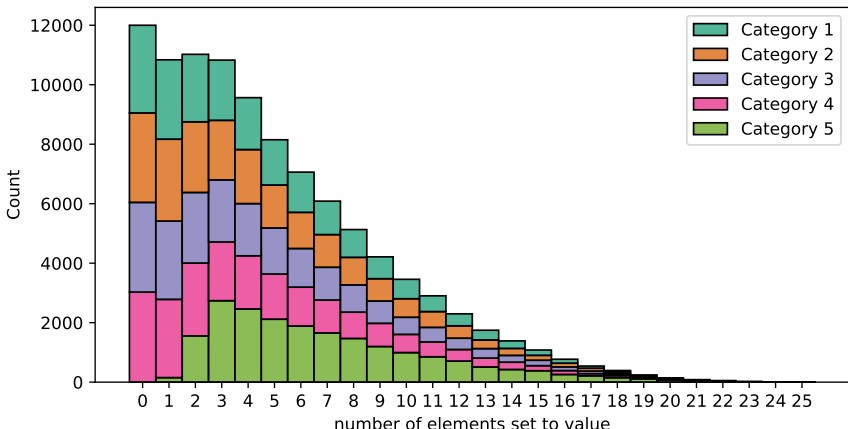

Figure 20: Histograms over the number of dictionary element entries set to each category for 20,000 repetitions of the sampling of dictionary elements for the `PestControl` benchmark. For each of the five categories and each value on the $x$-axis, the figure shows how often the number of entries in a dictionary element equals the value on the $x$-axis for the given category. For example, the count for $x = 0$ and category 5 is zero, indicating that all 20,000 dictionary points had at least one entry '5'. There is a considerably higher chance for a dictionary element entry to be set to category five than to any of the other categories.

Figure 20 shows five histograms over the number of dictionary elements set to each category. The values on the $x$-axis give the number of elements in a 25-dimensional categorical vector being set to a specific category. One would expect that the histograms have a similar shape regardless of the category. However, for category 5, we see that more elements are set to this category than for the other categories: The probability of $k$ elements being set to category 5 is almost twice as high as the probability of being set to another category for $k \geq 3$. In contrast, the probability that no element in the vector belongs to category 5 is virtually zero. This behavior is beneficial for the `PestControl` benchmark, which obtained the best value found during our experiments for $\boldsymbol{x}^* = (5, 5, \ldots, 5, 1)$

(see Section 4). While we see that the probability of each dictionary entry being set to category 5 is very low, we assume that we sample sufficiently many dictionary elements within a small Hamming distance to the optimizer such that `BODi`'s GP can use this information to find the optimizer.

The reason for oversampling of the last category lies in a rounding issue in sampling dictionary elements. In particular, for a given dictionary element $\boldsymbol{a}_i$ and a corresponding vector $\boldsymbol{\theta}$ with $|\boldsymbol{\theta}| = \tau$, for each $i \in \{1, 2, \ldots, \tau - 1\}$, Deshwal et al. [21] set $\lfloor D\boldsymbol{\theta}_i \rfloor$ elements to category $i$. The remaining $D - \sum_{i=1}^{\tau-1} \lfloor D\boldsymbol{\theta}_i \rfloor$ elements are then set to category $\tau$. This causes the last category to be overrepresented in the dictionary elements. For the choices of the `PestControl` benchmark, $D = 25$ and $\tau = 5$, the first four categories had a probability of $\approx 0.1805$. In contrast, the last one had a probability of $\approx 0.278$ for $10^8$ simulations[8]. We assume that the higher probability of the last category is the reason for the performance difference between the modified and the unmodified version of the `PestControl` benchmark.

## H.2 `COMBO` on categorical problems

On the categorical `PestControl` benchmark, `COMBO` [56] behaved similar to `BODi` [21]

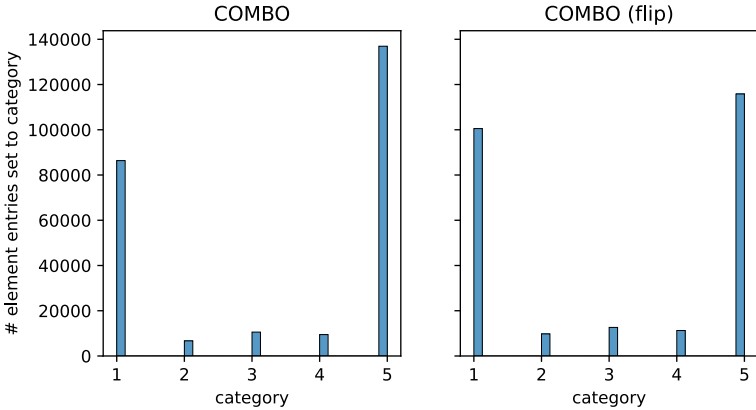

Figure 21: The histogram shows how many element entries of a 25-dimensional dictionary element are set to each of the five categories. There is a considerably higher chance for a dictionary element entry to be set to category 1 or 5 than to one of the other categories.

The histogram in Figure 21 shows how many element entries of a 25-dimensional dictionary element are set to each of the five categories. We see that the first and the last categories are over-represented on both benchmark versions. As discussed in Section 4, this benchmark attains the best observed value for $\boldsymbol{x}^* = (5, 5, \ldots, 5, 1)$. We observe that on the original and modified version of the benchmark, where the categories are shuffled, `COMBO` sets disproportionately many elements to categories one and five. Note that for the modified version of the benchmark, all categories are equally likely in the optimal solution.

We argue that this behavior is at least partially caused by implementation error in the construction of the adjacency matrix and the Laplacian for categorical problems[9]. This error causes categorical variables to be modeled like ordinal variables. According to Oh et al. [56], categorical variables are modeled as a complete graph (see Figure 22).

---

[8]The 95% confidence intervals for categories 1–5 are (0.1799, 0.1807), (0.1802, 0.1810), (0.1803, 0.1811), (0.1801, 0.1809), (0.2775, 0.2783). Pairwise Wilcoxon signed-rank tests between categories 1–4 and category 5 gives $p$ values of 0 ($W \approx 4.7 \cdot 10^{10}$ each).

[9]`https://github.com/QUVA-Lab/COMBO/blob/9529eabb86365ce3a2ca44fff08291a09a853ca2/COMBO/experiments/test_functions/multiple_categorical.py#L137`, last access: 2023-04-26

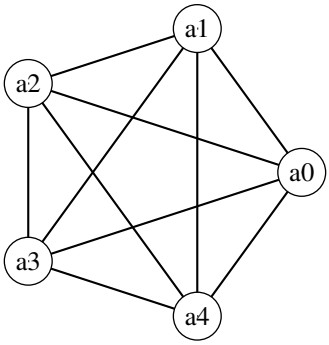

Figure 22: Following the description in the paper of Oh et al. [56], a categorical variable with five categories is modeled as a complete graph.

Looking into the source code of COMBO, we find the adjacency matrix for the first category of a categorical variable with five categories is constructed as

$$\begin{pmatrix} 0 & 1 & 0 & 0 & 0 \\ 1 & 0 & 1 & 0 & 0 \\ 0 & 1 & 0 & 1 & 0 \\ 0 & 0 & 1 & 0 & 1 \\ 0 & 0 & 0 & 1 & 0 \end{pmatrix},$$

which is the adjacency matrix for a five-vertex path graph.

