# OpenReview forum: "Bounce: Reliable High-Dimensional Bayesian Optimization for Combinatorial and Mixed Spaces"
_NeurIPS.cc/2023/Conference — NeurIPS 2023 poster_

### Official Review · Reviewer_FW4t · 2023-06-19

**Soundness:** 3 good
**Presentation:** 3 good
**Contribution:** 3 good
**Rating:** 7
**Confidence:** 4

**Summary:**

Most real-world optimization problems contain a mix of continuous, categorical, binary, and ordinal variables and might be high-dimensional. Although there are some methods that have investigated this problem, they are not always reliable with regard to finding a satisfactory optimum. In order to tackle this problem, the authors propose Bounce (Bayesian Optimization Using iNcreasingly high-dimensional Combinatorial and continuous Embeddings). This work seems to be a natural extension of BAxUS, which used a similar embedding strategy as for Bounce but only for continuous variables. It uses a novel trust region management system to grow or shrink the trust regions. The proposed method is tested on a representative range of test problems and compared with state-of-the-art benchmarks and shows convincing results.

**Strengths:**

- Most real-world optimization problems have mixed spaces and are high-dimensional for which vanilla Bayesian optimization methods are not suited, this work thus addresses an important problem.
- The proposed algorithm is tested on a broad range of problems and compared with state-of-the-art benchmarks and has a strong performance.
- The authors provide proof that the algorithm converges in the limit of infinite iterations, making it a reliable method.
- The paper is well-written and well-structured.

**Weaknesses:**

- In general, the paper is very complete and well-written and contains a very complete related work section. However, the paper does expect the reader to have a significant amount of knowledge of previous work. Especially about BAxUS and papers using TR management strategies. Perhaps the authors could provide a little bit more information and illustrations in this work to make it more easily accessible. For instance, provide an illustration of the binning procedures and/or about the TR management.

**Questions:**

- Although I do understand that this is a typical question in BO paper reviews, for most problems, benchmarks are performed for 200 iterations. Would you say that this is enough for the dimensionalities of the benchmark problems? For instance, for the 125-D MaxSAT, you would say you probably need a lot more data points for the surrogate model to model the objective function effectively. What are your thoughts on this and do you think some of the benchmarks can be better than Bounce at a higher budget?

- It is mentioned in Appendix C that the original implementations for COMBO, BODi and Casmopolitan are used. And that you use the same setting as in those works, what kind of settings are these exactly? Do these approaches use the same kernels for the GP models for example? Could this influence the performance on the benchmarks?

- Have the authors contacted Oh et al. regarding the bug in COMBO? If it is possible, it would of course be ideal if results using a bug-less version can be provided.

- I really like the results shown in Figure 5 regarding the efficacy of batch acquisition. In general, I like that the # batch evaluations are used on the x-axis as this directly shows what is good to use for users who are optimizing parallel processes. However, I'm also interested in how these plots look as a function of function evaluations. My intuition would say that b=1 is always better, as in this way the model is as informed as it can for the next iteration. Just out of interest, could you elaborate on this here or could you add a plot to the appendix that shows these results as a function of iterations?

- As a suggestion to your citations regarding chemical engineering and materials discovery. There also is a range of works regarding the optimization of lab equipment for sample analysis. See for instance [Hagan et al.](https://pubs.acs.org/doi/full/10.1021/ac049146x?casa_token=dnIitFW7lO4AAAAA%3Ado8InveDdcPw3TwtOVSQuRvM5NQhSFEo3M1jmpdpEHvRbzA0f1jYTJ2_bloYb7he8-Ofb8u96oMXvCvO), [Boelrijk et al.](https://scholar.google.nl/citations?view_op=view_citation&hl=nl&user=1z-BBwkAAAAJ&citation_for_view=1z-BBwkAAAAJ:9yKSN-GCB0IC). These problems can typically contain mixed spaces and many variables.

Some small textual remarks:
- Could it be that d_0 and d_{init} are interchangeably used? For instance lines 2 and 3 of Algorithm 1.
- Line 90, 'sequencies' should be 'sequences' or was this the originally proposed name by the BODi paper?
- Lines 277-278, as well should be as well as?
- Typo in Ass. 5. in Appendix Section a. One should be once?

**Limitations:**

The authors describe the societal impact of their work.
The authors do not describe the limitations of their work. Perhaps the authors could dedicate some sentences to this, for example, would the algorithm handle a noisy setting? Or does this violate the binning procedure to some extent?

---

> ### Author Rebuttal · Authors · 2023-08-09
>
> We appreciate the reviewer's remarks and will make sure they are appropriately addressed.
>
> > Perhaps the authors could provide a little bit more information and illustrations in this work to make it more easily accessible. For instance, provide an illustration of the binning procedures and/or about the TR management.
>
> Thank you for the suggestion! We will add a figure that explains the binning procedure and embedding to the camera-ready to facilitate the understanding of this crucial part of the algorithm.
>
> > Although I do understand that this is a typical question in BO paper reviews, for most problems, benchmarks are performed for 200 iterations. Would you say that this is enough for the dimensionalities of the benchmark problems?
>
> When running our experiments, we saw Bounce converging on most benchmarks after 200 function evaluations. A lower evaluation budget is common for discrete BO papers due to the increased cost of optimizing the acquisition function. For instance, the authors of BODi [1] worked with the same evaluation budget. However, we agree that running a higher evaluation budget would be interesting for the Labs and MaxSAT125 benchmarks where Bounce has not yet converged. Due to the high cost of these experiments, we will add these analyses to the camera-ready version.
>
> > It is mentioned in Appendix C that the original implementations for COMBO, BODi and Casmopolitan are used. [...] Do these approaches use the same kernels for the GP models for example? Could this influence the performance on the benchmarks?
>
> We agree that the choice of the kernel is a key aspect of these methods and is instrumental to their performance. Thus we use the exact settings and implementations of the respective authors. The core contributions of BODi and COMBO are their kernel constructions based on dictionary and diffusion kernels, respectively. We do not think replacing the kernel for these methods is reasonable.
>
> Note that Bounce uses the CoCaBO [2] kernel construction, also used in CASMOPOLITAN [3]. We will clarify this in the appendix.
>
> > Have the authors contacted Oh et al. regarding the bug in COMBO? If it is possible, it would of course be ideal if results using a bug-less version can be provided.
>
> We recently contacted the COMBO authors and are currently waiting for a response. We fixed the bug in COMBO independently, and the PDF uploaded with this rebuttal contains an updated figure for the PestControl benchmark, showing the original version of COMBO and a fixed version (marked with “(fixed)”). We observe that fixing this bug improves COMBO’s performance on this benchmark considerably so that COMBO outperforms BODi on the modified benchmark. As expected, fixing the bug makes COMBO agnostic towards the modification of the benchmark. Note that Bounce still outperforms all other algorithms. PestControl is the only benchmark in our experiments where COMBO is only affected by this bug (see Appendix D.2). We will update our discussion to reflect these new findings.
>
> > I really like the results shown in Figure 5 regarding the efficacy of batch acquisition. [...] However, I'm also interested in how these plots look as a function of function evaluations. [...] Just out of interest, could you elaborate on this here or could you add a plot to the appendix that shows these results as a function of iterations?
>
> A figure with the number of function evaluations on the x-axis for the batched version of Bounce is an interesting addition. As expected, large batches perform ‘worse’ when plotting against the number of function evaluations. There is almost no difference between small batches, highlighting the efficacy of our batching strategy. We added this figure to the PDF in the global response and will add it to the appendix in the paper.
>
> > As a suggestion to your citations regarding chemical engineering and materials discovery. [...]See for instance Hagan et al., Boelrijk et al.
>
> We agree on the relevance of this application area and will gladly add Hagan et al. and Boelrijk et al. as references for the practical applications.
>
> > Could it be that $d_0$ and $d_{\textrm{init}}$ are interchangeably used? For instance lines 2 and 3 of Algorithm 1.
>
> $d_0$ and $d_{\textrm{init}}$ are the same variable. We will revise this and keep only one of them.
>
> > Line 90, 'sequencies' should be 'sequences' or was this the originally proposed name by the BODi paper?
>
> The term ‘sequencies’ was introduced in the BODi paper [1] and describes the number of changes from 0 to 1 and vice versa in a bit vector — This is equivalent to ‘frequency’ over a continuous domain. We will clarify this.
>
> > The authors describe the societal impact of their work. The authors do not describe the limitations of their work. Perhaps the authors could dedicate some sentences to this, for example, would the algorithm handle a noisy setting? Or does this violate the binning procedure to some extent?
>
> While we discuss limitations in the discussion, we will expand this section, as suggested. We did not consider noisy function evaluations. In future work, we would like to explore the performance of a variant of Bounce on noisy problems where we replace EI or qEI with noisy EI or noisy qEI [4]. With these changes, we expect Bounce to also perform well on noisy problems.
>
> [1] Bayesian Optimization over High-Dimensional Combinatorial Spaces via Dictionary-based Embeddings, AISTATS, 2023 \
> [2] Bayesian Optimisation over Multiple Continuous and Categorical Inputs, ICML, 2020 \
> [3] Think Global and Act Local: Bayesian Optimisation for Categorical and Mixed Search Spaces, ICML, 2021 \
> [4] BoTorch: A Framework for Efficient Monte-Carlo Bayesian Optimization, NeurIPS, 2020

---

> > ### Comment · Reviewer_FW4t · 2023-08-14
> >
> > I have carefully read all reviewer comments and their respective rebuttals and I'd like to thank the authors for their hard work and effort. I am satisfied with the author's rebuttal and will keep my score as it is.

---

### Official Review · Reviewer_RWfy · 2023-07-06

**Soundness:** 3 good
**Presentation:** 3 good
**Contribution:** 3 good
**Rating:** 6
**Confidence:** 4

**Summary:**

The paper proposes a new BO method, namely Bounce, to tackle the problem of BO with combinatorial and mixed space. The key idea is based on the trust region approach (as with TurBO) and adaptive space (as with BAxUS), but is applied to the combinatorial and mixed variables. The proposed method is also enabled to work with batch. Experiments are conducted on various synthetic and real-world problems to evaluate the efficacy of the proposed method.

**Strengths:**

+ The paper’s writing is generally clear and easy to understand
+ The paper tackles an interesting problem, which is to solve BO problems with categorical and mixed variables
+ The methodology developed in the paper seems to be sound to me.
+ The experiments are conducted with various benchmark optimization problems (synthetic and real-world)


**Weaknesses:**

+ The proposed method seems to be largely an extension from existing methods: TurBO and BAxUS – normally this is alright if the results are impressive but there are some issues with the experiments which I will describe in more detail in the below bullets.
+ There is not much insight on why the proposed method, Bounce, works. There is a theoretical analysis in the appendix (Theorem 1) that shows the consistency of the Bounce. However, I think this theorem doesn’t have much meaning. It assumes that the search domain is finite, and the objective function is noiseless, and it states that Bounce will find a global optimum with probability of 1 when the number of samples N goes to infinity. But for a finite domain and noiseless observations, any algorithm will be able to find the global optimum?
+ One of the contributions of the paper is to conduct an in-depth analysis of two state-of-the-art algorithms for combinatorial BO, COMBO [45] and BODi [17], however, I don’t really find this to be interesting. Unless I missed it, it seems to be discussed in Section 4.6? But this is largely empirical and not much insight.
+ The experiments seem to lack of some well-known baselines for categorical and mixed search space, for example, SMAC, TPE, and some new BO methods like the work “Bayesian Optimization over Hybrid Spaces” by Deshwal et al (ICML 2021), and the work “Bayesian Optimization over Discrete and Mixed Spaces via Probabilistic Reparameterization” by Daulton et al (NeurIPS 2022).
+ Also, in the experiments, the number of iterations is quite small very high-dimensional problems. The number of iterations evaluated in all problems is just 200, which is too small.


**Questions:**

Please address my comments in the previous section (section Weakness)

**Limitations:**

I don’t find any dedicated section describing the limitations of the work.

---

> ### Author Rebuttal · Authors · 2023-08-09
>
> We are thankful for the reviewer's valuable input and will ensure their remarks are duly considered.
>
> > There is not much insight on why the proposed method, Bounce, works. There is a theoretical analysis in the appendix (Theorem 1) that shows the consistency of the Bounce. However, I think this theorem doesn’t have much meaning. It assumes that the search domain is finite, and the objective function is noiseless, and it states that Bounce will find a global optimum with probability of 1 when the number of samples N goes to infinity. But for a finite domain and noiseless observations, any algorithm will be able to find the global optimum?
>
> It is a crucial property of an algorithm to be consistent. We politely disagree with the assessment that any algorithm can find the optimum with an unlimited evaluation budget and a finite domain. In particular, most algorithms in the subspace BO literature (REMBO [1], HeSBO [2], Alebo [3], …) lack this property because they make a random bet on the embedding, and they are unable to recover from choosing a wrong embedding. See, for example, the discussion in [3].
>
> While we agree that regret bounds would be an interesting addition, no regret bounds have been proven for this line of work. Nevertheless, our design choices are well motivated, and the fact that Bounce first optimizes over a low-dimensional subspace and eventually reverts to optimization in the input space is crucial to its performance.
>
> > One of the contributions of the paper is to conduct an in-depth analysis of two state-of-the-art algorithms for combinatorial BO, COMBO [45] and BODi [17], however, I don’t really find this to be interesting. Unless I missed it, it seems to be discussed in Section 4.6? But this is largely empirical and not much insight.
>
> Regarding the deeper analysis of BODi and COMBO, we would like to refer the reviewer to Appendix D, which discusses the causes of the performance degradation of those algorithms in detail. In the camera-ready version of the manuscript, we will refer to this appendix more prominently in the main text —  We had to move this detailed analysis to the appendix due to space constraints.
>
> > The experiments seem to lack of some well-known baselines for categorical and mixed search space, for example, SMAC, TPE, and some new BO methods like the work “Bayesian Optimization over Hybrid Spaces” by Deshwal et al (ICML 2021), and the work “Bayesian Optimization over Discrete and Mixed Spaces via Probabilistic Reparameterization” by Daulton et al (NeurIPS 2022).
>
> We evaluated RDUCB [4], a recent additive method proposed at this year’s ICML, as an additional algorithm. We further added SMAC [5] to the comparison. Similarly to the CASMOPOLITAN [6] paper, we see SMAC performing poorly. We did not compare against probabilistic reparametrization (PR) [7] as we see PR falling in a different category, i.e., a meta-algorithm to optimize the acquisition function. However, we see the potential for improvement by combining Bounce with PR and would like to explore this in the future.
>
> > Also, in the experiments, the number of iterations is quite small very high-dimensional problems. The number of iterations evaluated in all problems is just 200, which is too small.
>
> We use the same evaluation budget as BODi [8], which also tackles high-dimensional problems, and Bounce converges after 200 iterations on most benchmarks. We want to emphasize that sample efficiency and the ability to find good solutions quickly are important properties of Bounce. To study the behavior for larger sample budgets, we increased the number of function evaluations to 500 for the Labs and ClusterExpansion benchmarks in the figures we submit with the rebuttal. We observe that Bounce continues to outperform the other algorithms and do not find a qualitative difference.
>
> > I don’t find any dedicated section describing the limitations of the work.
>
> For the camera-ready version, we will discuss limitations more prominently in the discussion section.
>
> [1] Bayesian optimization in a billion dimensions via random embeddings, JAIR, 2016 \
> [2] A Framework for Bayesian Optimization in Embedded Subspaces, ICML, 2019 \
> [3] Re-Examining Linear Embeddings for High-Dimensional Bayesian Optimization, NeurIPS, 2020 \
> [4] Are Random Decompositions all we need in High Dimensional Bayesian Optimisation? ICML, 2023 \
> [5] Sequential Model-Based Optimization for General Algorithm Configuration, LION, 2011 \
> [6] Think Global and Act Local: Bayesian Optimisation for Categorical and Mixed Search Spaces, ICML, 2021 \
> [7] Bayesian Optimization over Discrete and Mixed Spaces via Probabilistic Reparameterization, NeurIPS, 2022 \
> [8] Bayesian Optimization over High-Dimensional Combinatorial Spaces via Dictionary-based Embeddings, AISTATS, 2023

---

> > ### Comment · Reviewer_RWfy · 2023-08-17
> > **Thank you for the response**
> >
> > I would like to thank the authors for your response. The response has addressed many of my concerns so I decided to increase the score to 6. The reason I don't increas more is that I still have concerns regarding the very small number of iterations - normally I expect more for high-dim problems.

---

### Official Review · Reviewer_HPPy · 2023-07-06

**Soundness:** 3 good
**Presentation:** 2 fair
**Contribution:** 3 good
**Rating:** 7
**Confidence:** 4

**Summary:**

The paper proposes a new Bayesian optimization algorithm for combinatorial and mixed search spaces containing input variables of different types (continuous, binary, categorical, ordinal), which promises to be relevant for problems as varied as materials discovery, hardware design, neural architecture search, and portfolio optimization.
Bounce relies on a Gaussian process (GP) model of the objective function, which is based on lower-dimensional subspaces of the original search space, generated by partitioning input variables into “bins”.  Notably, the bins only contain input variables of the same type (e.g. continuous) and all variables in a bin are forced to take the same value during the optimization of the acquisition function, in effect operating in a lower-dimensional subspace. During the course of an optimization run, Bounce splits up bins into smaller ones, enabling the algorithm to propose candidates with increasingly finer structure. Bounce also leverages existing techniques for high-dimensional BO, including the trust-region approach.

**Strengths:**

- Strong empirical results, comparing against published baselines, in addition to ablations on the batch optimization performance.
- Performance is more robust to shifts of the solution than previously published methods.
- Sheds light on a non-trivial structural assumption of prior methods.
- Generally a well written paper.

**Weaknesses:**

- No theoretical analysis.
- Without a “modified” or “shifted” label for the left subplots of Figures 1, 2, 3, 4, the presentation is confusing and can easily be misread to suggest that the left subplots report results on the problems as defined in the literature, but they do not. Please add a “modified” or “shifted” label for the left subplots.

**Questions:**

- In the first 60 or so iterations, Bounce is outperformed by BODi on the non-modified 125d MaxSAT problem, as well as the  non-modified PestControl problem. Since sample-efficiency is a chief concern for BO methods, is there a way to leverage  BODi’s inductive biases for Bounce? On a high level, something like this might be possible since both methods leverage embeddings of the input space.
- Is there something more you can say about any theoretical aspects of the method? Does a global convergence guarantee hold? What can you say about the convergence rate, either local or global? How does the choice of embedding influence the convergence?
- Are there other embeddings approaches that could be combined with Bounce?

Suggestions:

- The description of the cardinal and ordinal embeddings are a bit repetitive and verbose, especially since they formally use the same approach. Can you unify the presentation, possibly include a single math-mode version of the embedding formula? That would aid the readability of the section.
- Line 191: “The embedding of ordinal variables follows [that of the] categorical variables”

**Limitations:**

Yes

---

> ### Author Rebuttal · Authors · 2023-08-09
>
> We appreciate the reviewer's insights and will now discuss their remarks.
>
> > Without a “modified” or “shifted” label for the left subplots of Figures 1, 2, 3, 4, the presentation is confusing and can easily be misread to suggest that the left subplots report results on the problems as defined in the literature, but they do not. Please add a “modified” or “shifted” label for the left subplots.
>
> We agree with their assessment that the Figure captions can be misleading when scanning the paper. We will add a label as you suggested. We submitted a PDF file with the rebuttal that shows the updated figures.
>
> > In the first 60 or so iterations, Bounce is outperformed by BODi on the non-modified 125d MaxSAT problem, as well as the non-modified PestControl problem. Since sample-efficiency is a chief concern for BO methods, is there a way to leverage BODi’s inductive biases for Bounce? On a high level, something like this might be possible since both methods leverage embeddings of the input space.
>
> Regarding the question about the inductive bias, we would like to stress that BODi is biased towards problems where the optimum lies in the origin or, for categorical problems, where the optimum is realized by setting all categories to the same ‘value’. In Appendix D, in the supplementary material submitted in April 2023, we discuss that most benchmarks in the BODi paper have this property due to their synthetic nature. We do not believe that this bias is reasonable for most practical applications.
>
> However, suppose the user has a prior belief that the optimal solution is located in or close to the origin or realized by setting all categorical variables to the same ‘value’. In that case, the user can similarly bias Bounce. We discuss this in Appendix B.3 in the supplementary material and show that this ‘low-sequency version’ of Bounce outperforms or is on par with BODi even on the version of the benchmark where the optimum is at the origin.
>
> > Is there something more you can say about any theoretical aspects of the method? Does a global convergence guarantee hold? What can you say about the convergence rate, either local or global? How does the choice of embedding influence the convergence?
>
> We refer to Appendix A in the supplementary material, where we prove that Bounce is consistent, i.e., converges to the global optimum in the limit. We did not prove regret bounds and would like to point out that no regret bounds are known for this line of work.
>
> > Are there other embeddings approaches that could be combined with Bounce?
>
> For the general case, we are not aware of any embedding that can generally be combined with Bounce. The Bounce embedding construction that conveys observations from a lower-dimensional to a higher-dimensional subspace is a key contribution of the paper. However, for the special case of purely continuous problems, Bounce could also use the HeSBO [1] embedding since it also uses a many-to-one mapping of input dimensions to target dimensions. Note that [2] showed that the HeSBO embedding has a lower worst-case probability of containing the optimum, so we opted for the BAxUS [2] embedding construction.
>
> > The description of the cardinal and ordinal embeddings are a bit repetitive and verbose, especially since they formally use the same approach. Can you unify the presentation, possibly include a single math-mode version of the embedding formula? That would aid the readability of the section.
>
> Thank you for the suggestion! We agree and will revise the section accordingly for the camera-ready version.
>
> [1] A Framework for Bayesian Optimization in Embedded Subspaces, ICML, 2019 \
> [2] Increasing the Scope as You Learn: Adaptive Bayesian Optimization in Nested Subspaces, NeurIPS, 2022

---

### Official Review · Reviewer_nHAf · 2023-07-07

**Soundness:** 4 excellent
**Presentation:** 4 excellent
**Contribution:** 3 good
**Rating:** 7
**Confidence:** 4

**Summary:**

The paper considers the problem of optimizing black-box functions defined over high-dimensional combinatorial and mixed continuous-combinatorial spaces. The key idea is to use the Bayesian optimization framework specialized to increasingly large nested embeddings of input dimensions in order to tackle the high-dimensionality challenge. The paper proposes separate embeddings for different class of input variables i.e. continuous, binary, categorical and ordinal. A trust region based approach is employed to enable parallel candidate evaluations in the proposed approach. Experiments are performed on multiple benchmarks to demonstrate the efficacy of the approach.

**Strengths:**

- The problem space considered in the paper is quite relevant and arises in multiple real-world applications. The paper is written really well describing the scope of the problem and associated applications.

- The proposed idea (although building on BAxUS and TurBO) is principled and explained well. The method clearly outperforms all state-of-the-art baselines on several benchmarks demonstrating its efficacy.

- The related work discussion and the amount of effort to compare with all the baselines in proper way is commendable and deserves credit.

Overall, I think the paper will be useful and practical contribution to this problem space of high dimensional combinatorial spaces.

**Weaknesses:**

Please see my question below.

**Questions:**

The intuition or principle behind the embedding specific to categorical variable is not entirely clear to me. While for continuous variables, the embedding comes from the idea of count-sketches, it is not clear immediately what is captured by assigning categorical variables of eve different cardinalities to the same bin. For example, if there is an outlier variable with very large cardinality or if the range of cardinalities is quite large, the embedding would be very sensitive to the outlier? Please expand the description about the categorical embedding more if possible.

---

> ### Author Rebuttal · Authors · 2023-08-09
>
> We're grateful for the reviewer's input and will be addressing their comments.
>
> We expanded the empirical section further and added RDUCB [1] and SMAC [2].
>
> > The intuition or principle behind the embedding specific to categorical variable is not entirely clear to me. While for continuous variables, the embedding comes from the idea of count-sketches, it is not clear immediately what is captured by assigning categorical variables of eve different cardinalities to the same bin. For example, if there is an outlier variable with very large cardinality or if the range of cardinalities is quite large, the embedding would be very sensitive to the outlier? Please expand the description about the categorical embedding more if possible.
>
> In the camera-ready version, we will add a visualization of the proposed binning procedure. We agree with the reviewer that binning variables of very different cardinalities can lead to undesired effects. Consider the case of ordinal variables. Representing a variable of low cardinality with a high number of ‘levels’ may cause the surrogate model to exhibit variability where there is none in the unknown black-box function: many levels correspond to the same function value since they are mapped to the same x-value of the low-cardinality variable. However, there is a sudden change between two levels if the next level corresponds to another x-value of the low-cardinality variable. Therefore, the surrogate model has wide flat regions with jumps between two levels. We have ideas to tackle this problem but left them for future work.
>
> We will discuss this limitation in the paper.
>
> [1] Are Random Decompositions all we need in High Dimensional Bayesian Optimisation?, ICML, 2023 \
> [2] Sequential Model-Based Optimization for General Algorithm Configuration, LION, 2011

---

> > ### Comment · Reviewer_nHAf · 2023-08-17
> > **Response to Rebuttal**
> >
> > Thank you for your time in responding to my queries. I am happy with the response and would like to keep my score of acceptance.

---

### Official Review · Reviewer_ACyR · 2023-07-08

**Soundness:** 3 good
**Presentation:** 3 good
**Contribution:** 3 good
**Rating:** 7
**Confidence:** 4

**Summary:**

This paper proposes an algorithm called Bounce, using nested embedding for mixed and combinatorial search space. Bounce partitions input variables into ‘bins’ and set all variables within the same bin to a single value to reduce the dimension. During the optimization, Bounce splits up bins into smaller bins for more refined optimization.

**Strengths:**

1/ The combinatorial and mixed optimization is significant and has wide range of applications in the real world.

2/ The algorithm is reasonable. Bounce designs a reasonable mapping for binary, categorical and ordinal variables, while a similar algorithm, BAXUS, which also uses nested random subspaces, has shown good performance on continuous optimization.


**Weaknesses:**

I don't have any major complaints. The proposed algorithm is a natural extension of BAXUS for categorical and mixed space. The experiments show that Bounce is reliable and can achieve good performance.

Here are some minor comments:

1/ One minor weakness is the the high-dimensional continuous spaces part in Section 2. A recent survey paper [1] categorizes these works into several categories, i.e., low-dimensional embeddings, decomposition, and variable selection. This paper mainly discuss the work in low-dimensional embeddings I think discuss the recent works in decompostion [3-4] and variable selection [5-6] would further strengthen this paper.

[1] A Survey on High-dimensional Gaussian Process Modeling with Application to Bayesian Optimization. ACM TELO, 2022.

[2] High-Dimensional Bayesian Optimization via Tree-Structured Additive Models. AAAI, 2021.

[3] Are Random Decompositions all we need in High Dimensional Bayesian Optimisation? ICML, 2023.

[4] Monte Carlo Tree Search based Variable Selection for High Dimensional Bayesian Optimization. NeurIPS, 2022.

[5] Fast and Scalable Spike and Slab Variable Selection in High-Dimensional Gaussian Processes. AISTATS, 2022.

2/ It would benificial to see the the disscussion (even experimental comparison) with a work with similar topic [6].

[6] Tree ensemble kernels for Bayesian optimization with known constraints over mixed-feature spaces

**Questions:**

I am puzzled about why the performance of BODi and COMBO can be sensitive to the location of the optima, which is equivalent to shuffling the labels of the categories of each variable. BODi uses Hamming distance and COMBO uses the combinatorial graph to model the relation of different labels for each variable. Different labels for the same variables are the same, so shuffling the labels will not impact the performance. Can you provide more explanation?

I think the variable selection methods [4-5] maybe also can be applied in the mixed spaces tasks. I want to see some discussions.

I'm glad to increase my score if you address the proposed isssues.

**Limitations:**

Yes.

---

> ### Author Rebuttal · Authors · 2023-08-09
>
> We thank the reviewer for their valuable remarks and are happy to address them.
>
> > 1/ One minor weakness is the the high-dimensional continuous spaces part in Section 2. A recent survey paper [1] categorizes these works into several categories, i.e., low-dimensional embeddings, decomposition, and variable selection. This paper mainly discuss the work in low-dimensional embeddings I think discuss the recent works in decompostion [3-4] and variable selection [5-6] would further strengthen this paper.
>
> We agree that comparing against decomposition-based techniques is a valuable addition to the paper. We, therefore, compare against the recent RDUCB [1] algorithm that relies on additive decompositions. Furthermore, we compare against the SMAC [2]  Random Forests surrogate model. We will further discuss the suggested papers in the related work section. The appendix submitted in April in the supplementary material contains an extended related work section that discusses Monte Carlo Tree Search (MCTS) based techniques [3, 4]. We plan to integrate this section into the main text if there is sufficient space and will expand the discussion on the recent work on decomposition and variable selection as suggested by the reviewer.
>
> > I am puzzled about why the performance of BODi and COMBO can be sensitive to the location of the optima, which is equivalent to shuffling the labels of the categories of each variable. BODi uses Hamming distance and COMBO uses the combinatorial graph to model the relation of different labels for each variable. Different labels for the same variables are the same, so shuffling the labels will not impact the performance. Can you provide more explanation?
>
> We hope Appendix D in the supplementary material answers the open questions about BODi and COMBO’s behavior. In the camera-ready, we will refer to this appendix more prominently in the main text since it contains important insights into these methods. Due to space constraints, we cannot move it to the main text.
>
> [1] Are Random Decompositions all we need in High Dimensional Bayesian Optimisation? ICML, 2023 \
> [2] Sequential Model-Based Optimization for General Algorithm Configuration, LION, 2011 \
> [3] Learning Search Space Partition for Black-box Optimization using Monte Carlo Tree Search, NeurIPS, 2020 \
> [4] Monte Carlo Tree Search based Variable Selection for High Dimensional Bayesian Optimization, NeurIPS, 2022

---

> > ### Comment · Reviewer_ACyR · 2023-08-16
> >
> > Thanks for your response. I will increase my score to 7.
> >
> > However, there is a question that may be missed. "it would benificial to see the the disscussion (even experimental comparison) with a work with similar topic [6] "
> >
> > [6] Tree ensemble kernels for Bayesian optimization with known constraints over mixed-feature spaces. NeurIPS, 2022.

---

> > > ### Author Response · Authors · 2023-08-19
> > >
> > > Thank you for your reply. We will add the suggested paper to the discussion and investigate whether an experimental comparison is feasible for the camera-ready version. We added RDUCB [1] and SMAC [2] to the comparison and were not able to run more algorithms in the time frame of the rebuttal.
> > >
> > > [1] Are Random Decompositions all we need in High Dimensional Bayesian Optimisation? ICML, 2023 \
> > > [2] Sequential Model-Based Optimization for General Algorithm Configuration, LION, 2011

---

### Author Rebuttal · Authors · 2023-08-09

We thank all the reviewers for their valuable feedback. We are pleased that the reviewers appreciated the problem's relevance, as acknowledged by reviewers ACyR, nHAf, and HPPy, along with their positive assessment of Bounce's performance across various benchmarks, as highlighted by reviewers ACyR, nHAf, HPPy, and FW4t.  We are happy to read that the reviewers, including reviewers nHAf, HPPy, and FW4t, found the paper to be well-written.

Several reviewers asked for a deeper analysis of the causes of the performance degradation we observed for BODi [1] and COMBO [2]. We want to point the reviewers to the in-depth analysis in Appendix D of the supplementary material that provides such an analysis — please notice that this year’s author guidelines do not allow the supplementary material to be submitted together with the main text.

Some reviewers asked for a comparison with additional methods. We submit a PDF file with the rebuttal that shows the updated figures. We added an evaluation of RDUCB [3], a recent algorithm presented at ICML 2023 that belongs to the class of additive methods. We also added a comparison with the well-known SMAC [4]. Bounce outperforms both the recent RDUCB algorithm and SMAC on all benchmarks.

Furthermore, we increased the number of function evaluations to 500 for the 50D-Labs and 125D-ClusterExpansion benchmarks. Due to the high computational cost, we could only run ten repetitions for BODi but will increase the number of repetitions for the camera-ready version. We don’t expect the results to change significantly since BODi has shown low variability in our experiments. Furthermore, not all 500 iterations have finished yet for COMBO due to its large running time. The plots show the mean averaged over  fifty runs. We observe that Bounce continues to outperform the other algorithms.

Based on the comment by reviewer FW4t, we fixed COMBO’s bug that affected its performance on the PestControl benchmark. Fig. 1 (c) in the PDF shows both the performance of COMBO with (marked “COMBO” in the legend) and without the bug (marked “COMBO (fixed)” in the legend). Fixing the bug improves COMBO’s performance on this benchmark and, as expected, makes COMBO agnostic towards the modification of the benchmark. Note that Bounce still outperforms COMBO and all other algorithms.

We note that we contacted the authors of BODi, and they acknowledged the sensitivity of BODi toward the location of the optimal point. We further contacted the authors of COMBO and are currently awaiting a response.


[1] Bayesian Optimization over High-Dimensional Combinatorial Spaces via Dictionary-based Embeddings, AISTATS, 2023 \
[2] Combinatorial Bayesian Optimization using the Graph Cartesian Product, NeurIPS, 2019 \
[3] Are Random Decompositions all we need in High Dimensional Bayesian Optimisation?, ICML, 2023 \
[4] Sequential Model-Based Optimization for General Algorithm Configuration, LION, 2011

---

### Decision · Program_Chairs · 2023-09-21

**Decision:**

Accept (poster)

**Comment:**

This paper considers the important problem of optimizing expensive black-box functions over high-dimensional combinatorial spaces with many real-world applications. The paper proposes an effective Bayesian optimization algorithm using a combination of ideas including BAxUS and trust-region methods to handle the high-dimensional challenge, and different embeddings for continuous, discrete, and ordinal variables. The paper showed good empirical performance on multiple benchmarks.

All the reviewers' were positive about this paper. They raised some clarification questions and the authors' rebuttal addressed many of them satisfactorily.

I recommend accepting the paper and strongly encourage the authors' to incorporate reviewers' comments and rebuttal discussion into the final paper.